# A comprehensive benchmark of single-cell Hi-C embedding tools

Dylan Plummer[1,2], Xiuyuan Lang [1,3], Shanshan Zhang[1,3], Yan Li[1], Jing Li [2,4] ✉ & Fulai Jin [1,2,4,5] ✉

Embedding is the key step in single-cell Hi-C (scHi-C) analysis which relies on capturing biological meaningful heterogeneity at various levels of genome architecture. To understand the strength and limitations of existing tools in various applications, here we use ten scHi-C datasets to benchmark thirteen embedding tools including *Va3DE*, a new convolutional neural network model that can accommodate large cell numbers. We built a software framework to decouple the preprocessing options of existing tools and found that no single tool works best across all datasets under default settings. The difficulty levels and preferred resolutions are different between benchmark datasets, and the choice of data representation and preprocessing strongly impact the embedding performance. Embedding cells from early embryonic stages relies on long-range compartment-scale contacts, but resolving cell cycle phases and complex tissue requires short-range loop-scale contacts. Both random-walk and inverse document frequency (IDF) transformation prefers long-range "compartment-scale" over short-range "loop-scale" embedding, while deep-learning methods better overcome sparsity at both scales and are more versatile with different resolutions. Finally, "diagonal integration" with independent data modal is a promising approach to distinguish similar cell subpopulations. Our findings underscore the significance of appropriate priors for scHi-C embedding and also offer insights into genome architecture heterogeneity.

Three-dimensional genome structure plays a critical role in distal gene regulation, and characterizing this structure across cell types and other biological states is of great interest in biomedical research. Chromatin conformation capture (3C)[1–3] technologies, including Hi-C[4] can identify distal chromatin interactions and profile 3D genome architecture. More recently, various single-cell Hi-C technologies, including single-cell Hi-C, sci-Hi-C, sn-m3C-seq/scMethyl-HiC, Dip-C[5–9], have been developed to map the heterogeneity of genome architectures in the context of cell cycle, early development, and complex tissues.

Several dimension reduction techniques, including deep representation learning methods, have been applied to embed scHi-C data. The published pipelines are vastly different regarding recommended resolutions, data representations, and preprocessing procedures. However, these settings are known to prefer 3D genome features at different scales. Specifically, bulk Hi-C analyses from low- to high-resolution reveals a hierarchy of multi-scale 3D genome features, including compartment (~500 kb–1 Mb resolution)[4], topologically associating domains (TADs) (~50 kb resolution)[10,11], and chromatin loops (kb resolution)[12,13]. Since change of cell state may involve

[1]Department of Genetics and Genome Sciences, School of Medicine, Case Western Reserve University, Cleveland, OH, USA. [2]Department of Computer and Data Sciences, Case Western Reserve University, Cleveland, OH, USA. [3]The Biomedical Sciences Training Program (BSTP), School of Medicine, Case Western Reserve University, Cleveland, OH, USA. [4]Department of Population and Quantitative Health Sciences, Case Western Reserve University, Cleveland, OH, USA. [5]Case Comprehensive Cancer Center, Case Western Reserve University, Cleveland, OH, USA. ✉e-mail: jingli@case.edu; fxj45@case.edu

alteration of genome architecture at different scales, one embedding pipeline at one resolution is unlikely to be adequate for all applications. It is therefore necessary to benchmark the performance of existing pipelines at different resolutions in various biological settings (Supplementary Fig. 1).

The computational challenge unique to scHi-C embedding is to overcome severe data sparsity while capturing state-specific genome architecture. Firstly, Hi-C sparsity has a strong impact on the recognition of high-resolution TAD- or loop-level structures because the data sparsity grows quadratically with increasing resolution. Secondly, Hi-C sparsity also impacts the recognition of low-resolution compartmental structures because compartmental interactions are long-range (>1 Mb), and data sparsity of Hi-C contact maps is much worse at long-range than at mid- or short-range. All embedding tools must make decisions with data representations and preprocessing steps to address the issues of high-dimensionality and data sparsity explicitly or implicitly. It remains unclear how these decisions impact the embedding performance in various biological applications. Here, we develop a software suite that unifies multiple tools after decoupling their preprocessing options, then benchmark their performance in various applications under different settings. This tool can help users explore optimal priors for scHi-C data embedding and to develop novel embedding strategies.

## Results

### Single cell Hi-C datasets and embedding tools for benchmark analysis

We performed a systematic benchmark analysis of thirteen unique embedding pipelines with ten published scHi-C datasets from various protocols (Fig. 1a, b and Supplementary Tables 1–2). The data analyzed consists of six mouse datasets and four human datasets, with the main biological settings distinguished as early embryogenesis (oocyte-to-zygote transition[14] and preimplantation embryos[15]), complex tissue (e.g., mouse/human brain)[5,16–18], cell cycle[6,19], and synthetic mixtures of multiple cell lines[7,20] (Fig. 1a, Supplementary Table 1). For all benchmark datasets, the original studies have determined the identity of every cell using reliable orthogonal approaches or co-assays (Fig. 1a and Supplementary Table 1). We therefore use the published cell identity information as the ground truth. The most complex dataset is the human brain snm3c-seq dataset with >32 K cells[17] including 29 cell populations, 22 of which are different neuron subtypes (Fig. 1a). The thirteen embedding pipelines include nine published tools including: (i) six "conventional" pipelines relying on matrix or tensor decomposition of the contact matrices (scHiCluster[21], fastHiCRep[22], InnerProduct[23], SnapATAC2[24], cisTopic[20], and Fast-Higashi[25]); (ii) scGAD[26] as a method with biological prior which explicitly models gene body-associated interactions for single cell embedding; (iii) two deep learning methods (Higashi[27], scVI-3D[28]) which utilize deep neural networks (NNs) (Fig. 1b and Supplementary Table 2).

We also developed four additional in-house pipelines and included them into benchmark analysis (Fig. 1b and Supplementary Table 2) for the following reasons. Firstly, we included a 1D-PCA method to provide an overall baseline for all embedding tools. This simplistic baseline method ignores 2D data structure by compressing Hi-C matrices into one-dimensional vectors; the entries of each vector represent the total cis-contacts of all loci in the genome (equivalent to the raw "cis-visibility"[29]) (**Methods**). Secondly, we considered the possibility of using TAD features to embed cells and developed two single cell embedding pipelines which identify TAD boundaries using two different methods (InsScore[30], deTOKI[31]), followed by PCA embedding (**Methods**). Thirdly, adding to the repertoire of deep NN-based scHi-C embedding methods, we also developed a method named Va3DE tool for comparison. Va3DE adapts the Variational Deep Embedding (VaDE) method[32] to embed scHi-C data by using convolutional layers to encode the single cell contact maps (**Methods**). The

motivation to include Va3DE into the benchmark is to compare the classic convolutional neural network to the hypergraph used by Higashi[27] and the distance-strata used by scVI-3D[28] for Hi-C data representation. Another motivation to include Va3DE is to assess how deep embedding methods perform at high-resolution, since Higashi and scVI-3D are typically only used at low resolution (500 kb) due to memory limits. Va3DE is less memory demanding because it processes single cells data in batches while still scaling to the entire dataset (**Methods**).

### Embedding tools show variable performance across distinct types of applications

A good embedding method should place cells from the same population near one another in the embedding space. We can therefore assess the quality of embeddings by examining if it yields clusters that agree with annotated cell types. In this study, we primarily use unsupervised K-means for clustering (see Additional Discussions about clustering methods in **Methods**) and use adjusted rand index (ARI)[33] to quantify the clustering performance of each embedding tool in all benchmark applications. ARI takes in a predicted clustering assignment and measures if cells from the same cell type belong to the same cluster, with a correction for cluster sizes. We also report two other clustering metrics, including the normalized mutual information (NMI) and cell type average silhouette scores (ASW) (see **Methods**). For every test, we also computed a cumulative AvgBIO score by averaging the ARI, NMI, and ASW metrics.

To obtain an initial rank, we assessed three resolutions (1 Mb, 500 kb, 200 kb) for all applications, and observed that a substantial variation of ARIs as resolution changes (Supplementary Fig. 2). To better summarize these results, we chose the resolution with best ARI scores for each application and summarized the corresponding ARI (Fig. 1c) and AvgBIO scores (Fig. 1d); the chosen resolutions are also indicated in Fig. 1d. We finally ranked all embedding tools by their median AvgBio rank across all ten datasets (Fig. 1d). We also provided an estimation of the runtime for each method with two typical cell numbers (1.8 K and 18 K cells) at different resolutions (Fig. 1e, also see additional discussions in **Methods**). We noted that several methods have difficulties achieving 200 kb resolution due to memory limits, especially when the cell number is high (Fig. 1e and Supplementary Fig. 2). Surprisingly, fast-Higashi is more memory demanding than Higashi during the model initiation step (Fig. 1e); the memory performance of fast-Higashi can be improved by disabling its default random-walk step, noting that fast-Higashi recommend disabling random-walk only for "high quality scHi-C data". In this study, we keep random-walk option for consistency. Additionally, we also visually examined the embeddings using t-SNE or UMAP for qualitative comparisons of method performance (Supplementary Fig. 3).

Two deep-learning methods (Higashi and Va3DE) have the best scores, followed by SnapATAC2, which has comparable score and much less computational burden (Fig. 1d, e). Three other "conventional" methods, Fast-Higashi, InnerProduct, and scHiCluster, also achieve solid performance in most applications (Fig. 1c–e and Supplementary Fig. 3). However, even the top performers may show variations across different applications. For example, Higashi performs well across multiple applications but still ranks low with preimplantation embryos (Fig. 1c). In contrast, scHiCluster is among the best tools with the two embryogenesis datasets but works less well with cell cycle and complex tissue datasets (Fig. 1c). Methods using TAD prior (InsScore and deTOKI) work poorly and rank even lower than the 1D-PCA baseline, suggesting that TAD is not an informative feature for embedding (Fig. 1d). Gene prior (scGAD) can distinguish different cell types in synthetic mixtures or complex tissues but does not work well with cell cycle or early embryo stages, most likely because it only uses gene body contacts for embedding (Fig. 1c). There also seems to be a trend that early tools work particularly well with the datasets that they

were initially evaluated on (listed in Supplementary Table 2). For example, *cisTopic* performs best with sciHi-C mixtures; *scHiCluster* excels at the mouse oocyte and zygote dataset; *InnerProduct* shows the best circular pattern with the mESC cell cycle data (Fig. 1c, d and Supplementary Fig. 3). All these observations indicate that scHi-C embedding is not a winner-takes-all problem.

The difficulty levels and preferred resolutions are different between benchmark datasets. (i) HiRES and CARE-seq data are hard to embed without the use of co-assayed scRNA-seq data: they showed lower ARI and AvgBIO scores than all other datasets, regardless of the tools used, perhaps due to compromised Hi-C data quality (Fig. 1c, d and Supplementary Figs. 2–3, see Additional Discussions in **Methods**). (ii) The human brain atlas data showed lower ARI scores than the other two complex tissue datasets (Fig. 1c, d and Supplementary Fig. 2) due to the difficulty to distinguish the 22 neuron subtypes (out of 29 total annotated cell populations) in this dataset. (iii) Synthetic mixture datasets prefer the lowest resolution at 1 Mb regardless of the embedding methods (Fig. 1d), and the baseline 1D-PCA surprisingly outperforms all other tools (Fig. 1c). This is probably because the cell lines used in the artificial mixtures are so different that low-resolution 3D genome features can still easily distinguish them. (iv) The two embryogenesis datasets also show a preference toward low-resolution at least for some tools, but cell cycle and complex tissue datasets prefer higher resolution at 500 kb or 200 kb (Fig. 1d).

To understand why each tool succeeds or fails in different applications, we setup a software package named *SCORE* (Single-cell Chromatin Organization Representation and Embedding) to decouple the preprocessing options from the embedding algorithms, thus comprehensively compared all possible "mix-and-match" options allowed by every pipeline (Supplementary Table 2). The options include: (i) choice of different resolutions; (ii) size range of chromatin contacts; (iii) optional preprocessing such as random walks and inverse document frequency (IDF) transformation. We also evaluated the impacts of read-depth on embedding performance.

### Early embryogenesis and complex tissue data prefer different loop sizes for embedding

The oocyte-to-zygote transition[14] and preimplantation embryo[15] datasets are challenging for many pipelines, including the deep-learning-based tools. Interestingly, as mentioned above, we often achieve the top scores on these datasets at low resolution of 1 Mb or 500 kb (Fig. 1d and Supplementary Figs. 2–3). The 3D genome architecture undergoes extensive reprograming after fertilization, particularly the high-order genome architectures, including compartments and TADs. These Mb-scale structures are missing or weakened in oocytes and zygotes and will be gradually re-established during the first few cell divisions of early embryogenesis in an allele-specific manner[14,15,34–37]. We therefore suspect that long-range contacts might be particularly important to embed the early embryogenesis data.

To test this hypothesis, we limit the size of contacts used for embedding and investigate how the size range affects the embedding performance. Indeed, for both *InnerProduct* and *SnapATAC2*, the separation of oocytes from zygotic nuclei requires long-range contacts >2 Mb, and we can achieve good separation of cell populations even if we only use the >2 Mb contacts (Fig. 2a, b, quantitation in Fig. 2e). Similarly, preimplantation embryo data also require long-range contacts to fully separate the early embryonic stages (Fig. 2c, d, quantitation in Fig. 2e). We also evaluated the distance effects with other tools including *cisTopic, scHiCluster,* and *Va3DE* and observed the same trend (Fig. 2e and Supplementary Fig. 4). These conclusions are also Supplementaryorted by a visualization of the per-strata inner products between cells from each cell stage. We see ultra-large genome structures in the paternal zygotes but less so in maternal zygotes, while oocytes have heterogenous genome structure at various size scales (Fig. 2f, **Methods**). For the preimplantation embryo dataset, we see

clear 3D genome reprograming from 1C to 64C. Particularly, 2C heterogeneity emerges between 4–6 Mb, and 4c and 8c heterogeneity emerges from 4 to 10 Mb. These results are also consistent with a whole chromosome visualization in which we observe a linearly compressed chromatin architecture in oocytes and zygotes and the gain of more high-order structures as the embryos continue to develop[36] (Supplementary Fig. 5a). When embedding cells from the two independent datasets together, we can observe a trajectory of early embryogenesis in which zygotes and 1C cells show highest similarity (Supplementary Fig. 5b).

In contrast, we show in Fig. 3 that human prefrontal cortex and mouse hippocampus datasets both rely on the short-range contacts for best embedding. For both *InnerProduct* and *scHiCluster*, optimal ARI are achievable when only using contacts <2 Mb (Fig. 3a, d, left panels). Contacts >2 Mb can still distinguish non-neuronal cell types but cannot separate neuronal subtypes in either human PFC (8 neuronal subtypes) or mouse hippocampus (5 neuronal subtypes) datasets (Fig. 3b, e). To quantify this observation, we isolate the neuronal populations from the same embeddings and re-cluster them to compute a "neuron ARI" (Fig. 3c, f). We see no difference between neuron ARI when running *InnerProduct* using contacts <20 Mb compared to <2 Mb, indicating that including larger contacts does not further improve embedding. Interestingly, when running *scHiCluster*, inclusion of >2 Mb contacts lowers the neuron ARI (Fig. 3a, d, compare the left and middle panels, quantitation in Fig. 3c, f). This observation suggests that when both short- and long-range contacts are included in the embedding, *scHiCluster* pays more attention to long-range interactions (due to its random-walk process, will discuss later), which hurts the neuron ARI.

### Short-range contacts can distinguish G1-S cell cycle stages except M-phase

*InnerProduct* and several other tools can generate a circular pattern with the cell cycle data (Supplementary Fig. 3)[22,23,38]. We further find that when applying *InnerProduct*, embedding with short-range (<200 kb, less so for <2 Mb) interactions Supplementaryresses the mitotic cells and only shows a circular pattern of G1-S stages (Fig. 4a, compare the top and bottom row); inclusion of longer-range interactions obscure the distinctions between G1-S phases but separate a small number of mitotic cells (Fig. 4a, b). This is consistent with the original scHi-C study that G1-to-S phase transition is characterized by increasing fraction of local contacts (<2 Mb) and decreasing fraction of mitotic contacts (>2 Mb, <12 Mb) (Fig. 4d)[6]. The visualization of per-strata InnerProduct heatmap confirms that the four G1-S phases show heterogeneity primarily only within the first few strata (Supplementary Fig. 6). Furthermore, our whole-chromosome visualization also confirms that metaphase is characterized by a linearly compressed chromatin with larger contact size (Fig. 4c and Supplementary Fig. 6)[39–41].

Interestingly, when using *InnerProduct* to embed this dataset at 200 kb resolution, we observed a circular profile from the 64-cell embryos, which is reminiscent of the cell cycle pattern (Fig. 4e, f). Indeed, the circular pattern of 64-cell embryos is remarkably like the cell cycle data when we plot the fraction of short-range contacts against mitotic contacts (compare Fig. 4d, e). When we embed the 64-cell embryo data on its own, cells from 64 C embryos form a circle like the G1-S cell cycle plot (Fig. 4f). We therefore conclude that the *InnerProduct* embedding of early embryo data contains both long-range 3D genome heterogeneity representing developmental stages and short-range 3D genome heterogeneity representing cell cycle stages.

### Random walk and IDF transformation are both biased towards long-range contacts

*ScHiCluster* was originally designed and benchmarked on the oocyte/zygote dataset where random-walk was shown to improve the

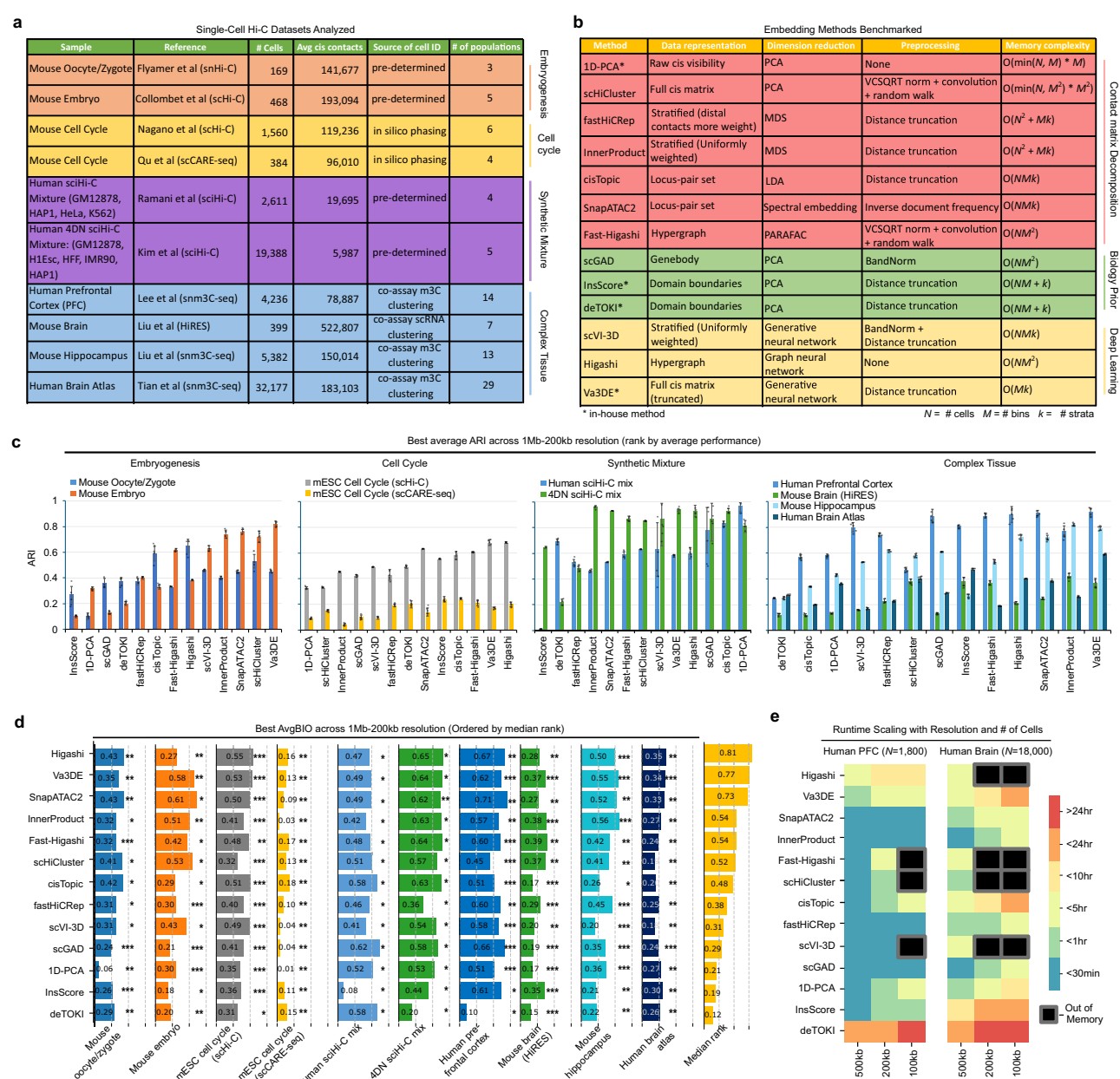

**Fig. 1 | Overview of scHi-C clustering benchmark. a** Summary of 10 scHi-C datasets colored by categories of biological questions. **b** Summary of the 13 benchmarked embedding pipelines colored by classification. Red: methods relying on decomposition of the single-cell contact maps or decomposition of a cell-cell similarity/covariance matrix; green: methods approaching the problem from a biological perspective, extracting known features a priori such as gene bodies and TADs; yellow: deep learning-based methods. Most methods operate directly on the raw count data with minimal preprocessing, except extracting a limited number of contact matrix strata to limit the interaction distances considered. *scHiCluster* applies linear convolution and random-walk imputations to each contact matrix. SnapATAC2 transforms counts based on inverse-document-frequency transformation (IDF). *scGAD* and *scVI-3D* transform matrix counts into z-scores using the *BandNorm* normalization. The memory requirements of methods scale according to the number of cells (N), the selected resolution, which determines the bin size and number of bins (M), and for distance stratified methods the number of strata to

analyze (k) (see Methods). Note *Va3DE* is the only method with no dependency on the number of cells. **c** Best ARI (adjusted rand index) across three resolutions used (200 kb, 500 kb, 1 Mb) on four categories of scHi-C datasets: embryogenesis, mESC cell cycle, synthetic cell line mixture, and complex tissue. Methods are ordered by average rank across each category. *n* = 5 runs for each method on each dataset at each resolution. Data are presented as mean values ± SEM. **d** Bar plots: the first 10 columns show the best AvgBIO scores for all embedding applications among three resolutions (200 kb=***, 500 kb=**, 1 Mb=*); the chosen resolutions are indicated with stars. The rightmost bar plot ranks the embedding methods based on a median rank metric. Based on the AvgBIO scores, we firstly compute the percentile ranks (ranging 0 -1, 1 being the top performing method) of all methods when applied to the same dataset. Each method will then get a rank score as its median rank across all 10 datasets. **e** Summary of runtime analysis under situations (1.8 k to 18 k) as we increase resolution. Some methods are unable to run due to memory constraints at higher resolutions or high cell count. Source data are provided as a Source data file.

embedding[21]. We confirmed that *scHiCluster* is a top performer with the two early embryogenesis datasets (Fig. 1c). Random-walk considers Hi-C data as an undirected graph and imputes edges in favor of densely connected nodes which represent large interacting neighborhoods, such as TAD and compartment structure (Supplementary Fig. 7a).

Removing random-walking from scHiCluster significantly compromises the embedding of both early embryogenesis datasets (first row of Fig. 5a, b, quantitation in Fig. 5c). To further explore the impact of random-walking imputation, we add this step to other embedding tools and find that the modified *fastHiCRep* and *InnerProduct* indeed

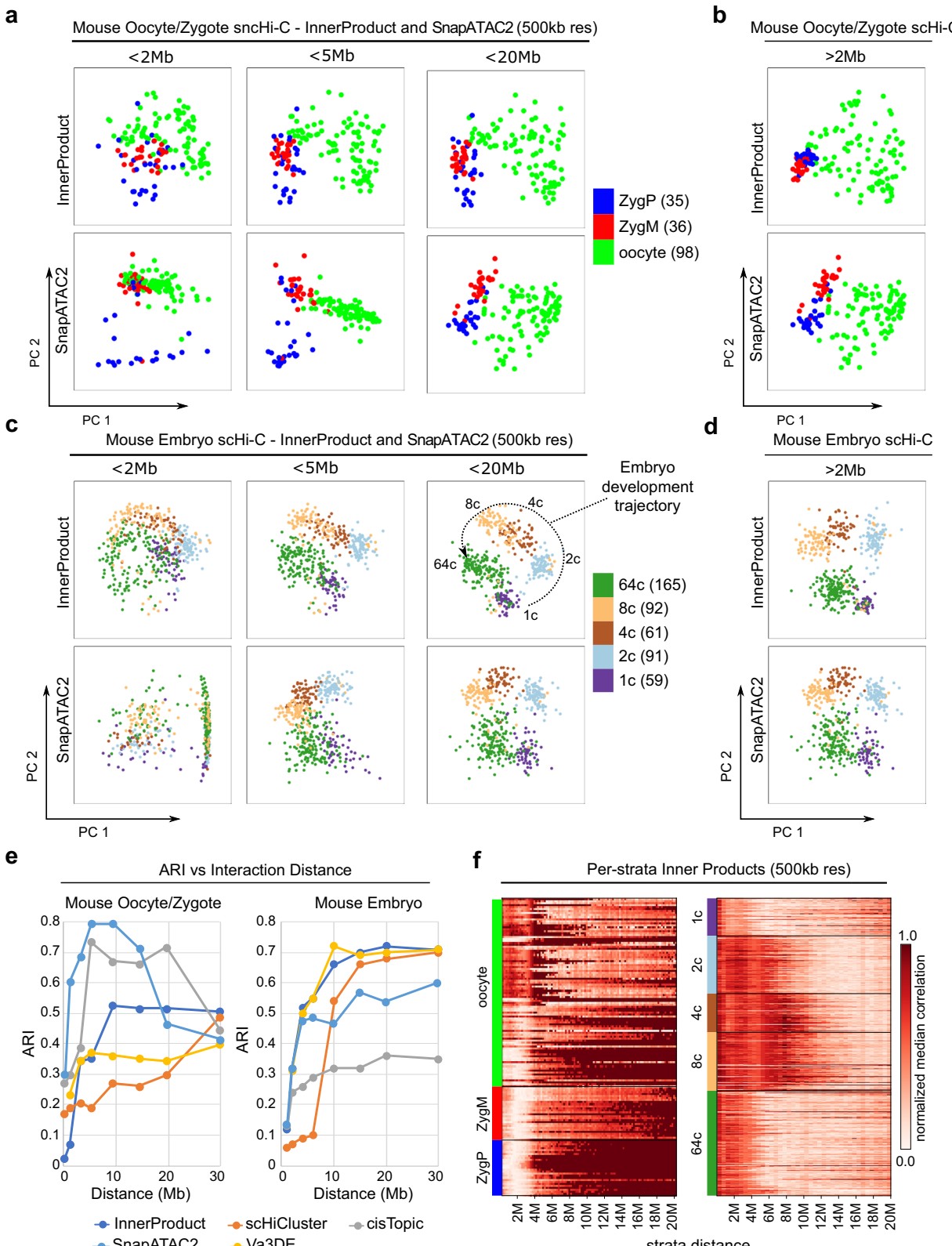

**Fig. 2 | Large contacts distinguish early embryogenesis cell populations.**
**a** Mouse oocyte/zygote scHi-C dataset embedded using *InnerProduct* and *Va3DE* at 500 kb resolution with varying maximum interaction ranges. **b** Mouse oocyte/zygote embedding after ignoring interactions within 2 Mb. **c** Mouse early embryo cells embedded using *InnerProduct* and *Va3DE* at 500 kb resolution with varying maximum interaction ranges. **d** Mouse early embryo cells embedded after ignoring interactions within 2 Mb. **e** Plot ARI against max interaction distance considered for the two mouse embryogenesis datasets. **f** Aggregated inner product values at each stratum of the mouse embryogenesis datasets. Here we run the *InnerProduct* method and visualize the normalized median inner product of each cell at various distance ranges. High values indicate high per-strata similarity occurring at this range. Source data are provided as a Source data file.

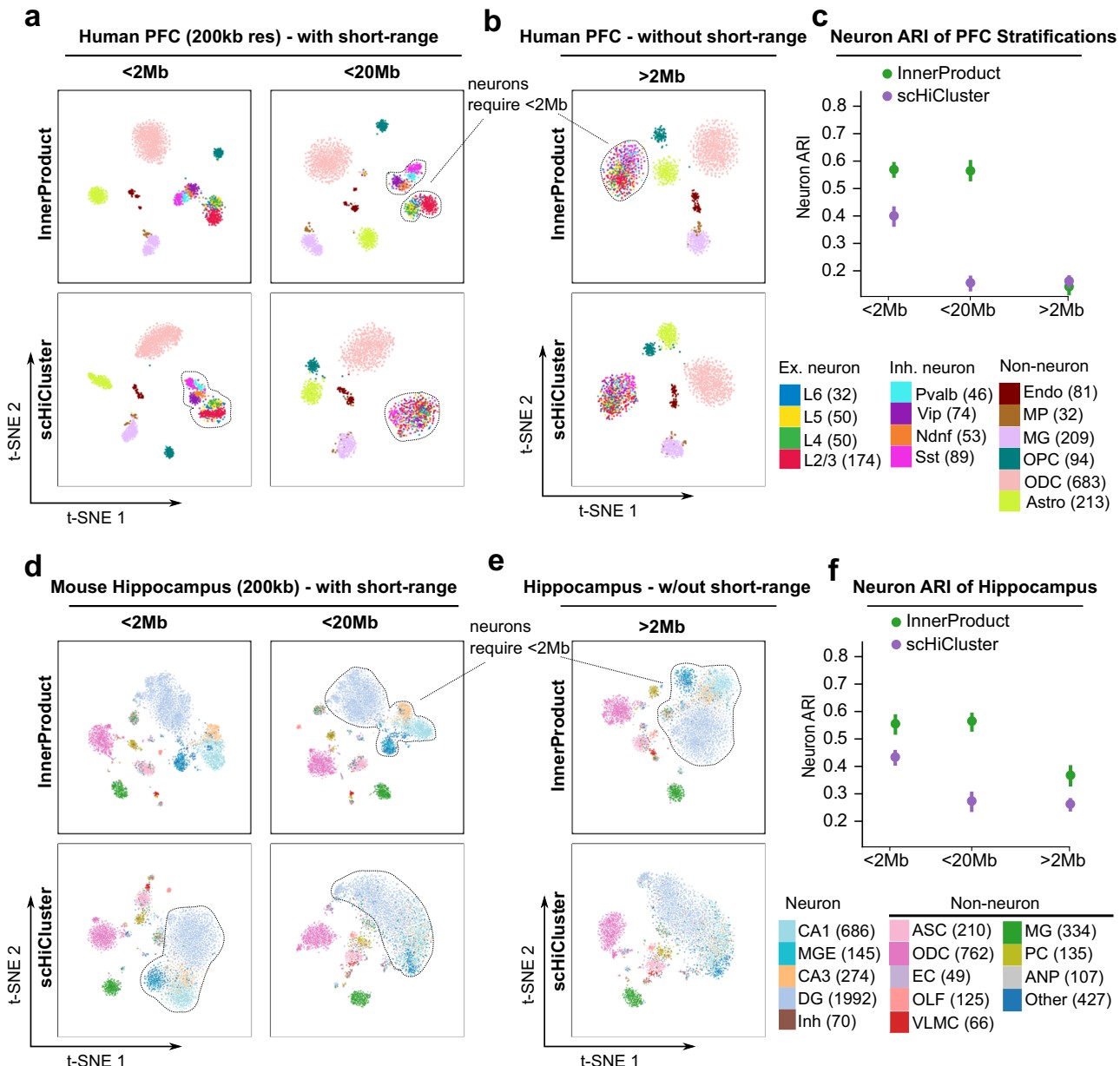

**Fig. 3 | Sub-Mb contacts best distinguish cell populations in complex tissues.**
**a** Human prefrontal cortex snm3c-seq data embedded using *InnerProduct and scHiCluster* at 200 kb resolution with varying maximum interaction ranges.
**b** *InnerProduct and scHiCluster* can still distinguish the non-neuron populations after excluding the contacts within 2 Mb. **c** Compare ARI from **a**, **b** only considering the neuron populations (Neuron ARI). **d**–**f** Same benchmark analysis as in (**a**–**c**) with the mouse hippocampus dataset. *n* = 5 runs for each method. Data are presented as mean values ± SEM. Source data are provided as a Source data file.

achieve better embedding performance on the two early embryogenesis datasets(Supplementary Fig 8a–c). We mentioned above that the deep-learning tool *Higashi* ranks low with the mouse embryo data (Fig. 1c) and strikingly, additional random-walk fixed this problem for *Higashi* (second row of Fig. 5a–c). The only exception is *Va3DE*, which already performs well on these datasets, suggesting that *Va3DE* learns high order structures without the need to explicitly guide the embedding towards this prior (third row of Fig. 5a–c). It worth mentioning that scHiCluster uses an iterative random-walk till a convergence threshold is met[21], this is necessary as too much walk hurts the embedding even for the mouse embryo data (Supplementary Fig. 7b–e).

Because scHiCluster ranks low with cell cycle data (Fig. 1c) and has difficulty distinguishing neuron subtypes in complex brain tissues when we include long-range contacts into embedding (Fig. 3), we

suspect that random walk may negatively impact the embedding of short-range contacts. Indeed, although both *InnerProduct* and *fastHiCRep* can embed cells in G1-S phases into a full circle in PC space, random-walk compresses the circular profile, and only a small number of M-phase cells stand out (Fig. 5d). Random-walk is also detrimental to the clustering of neuron subtypes in complex brain tissues. Specifically, when we use *InnerProduct* or *fastHiCRep* to embed the human PFC or mouse hippocampus snm3C-seq dataset, the neurons subpopulations are less separated if random-walk step is performed prior to the embedding (Fig. 5e, f, quantitation of Neuron ARI in Fig. 5g), regardless of the random-walk parameters (Supplementary Fig. 7d–e). The separation between neurons and non-neurons is unaffected. This is because neuron cell types (such as DG, CA1, and CA3 in mouse hippocampus data) have highly similar compartment structures, but the compartment of non-neuron cell types (such as ODC and MG) are

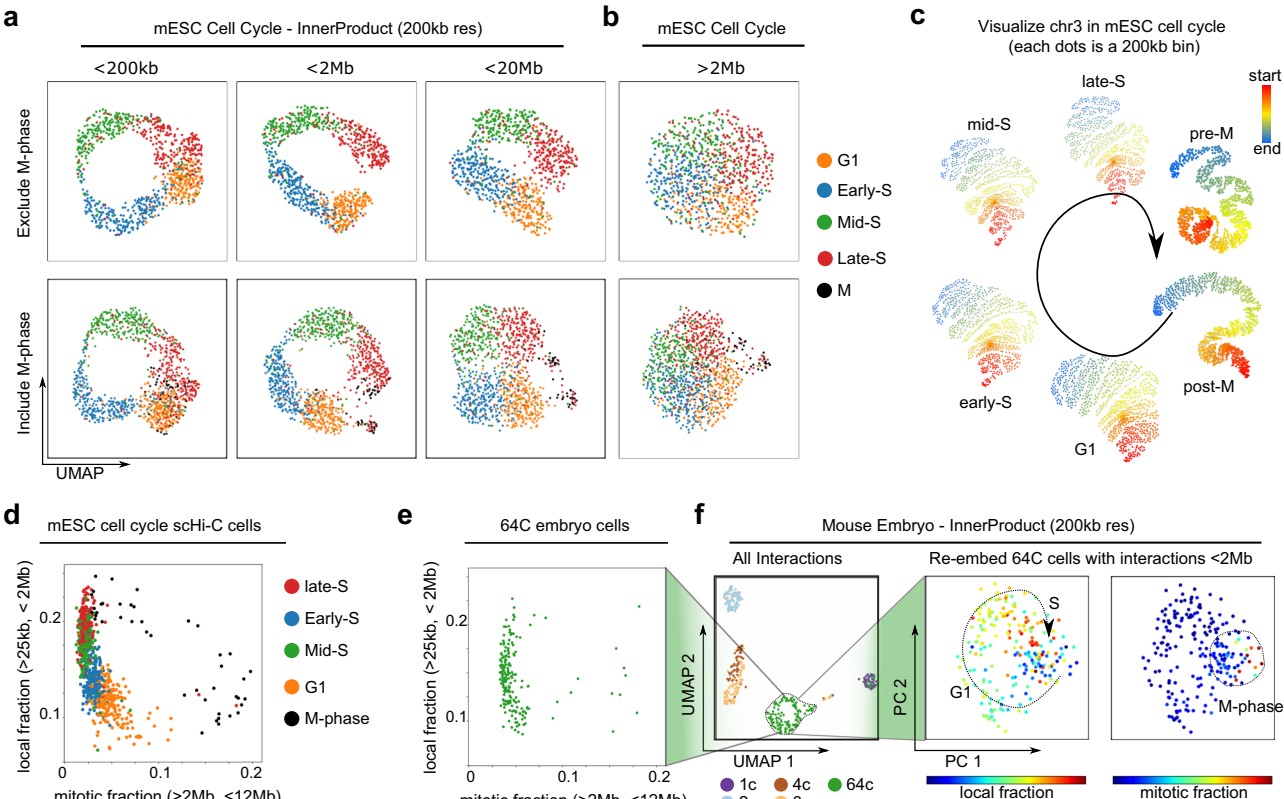

**Fig. 4 | Short-range contacts encode G1-S phases into a circle except M-phase in cell cycle. a** Cell cycle scHi-C contacts embedded using *InnerProduct* at 200 kb resolution with varying maximum interaction ranges. Top row: M-phase cells are excluded; bottom row: M-phase cells are included. **b** Embedding cell cycle data with interactions >2 Mb loses G-S cyclic dynamics but maintains M-phase heterogeneity. **c** Full chromosome graph visualization of cell cycle phases (**Methods**). **d** Cell cycle

scHi-C cells are plotted by the fractions of local contacts (>25 kb, <2 Mb) and long-range mitotic contacts (>2 Mb, <12 Mb). **e** Plot the 64C stage cells from the mouse embryo dataset in the same way as in (**d**). **f** Left: 64C embryo cells form a circle when embedding full dataset with *InnerProduct* at 200 kb resolution. Middle and right: embedding only the 64C stage cells using interactions within 2 Mb. Cells are colored based on the fractions of local (middle) and mitotic contacts (right).

clearly different (Fig. 5h). Since a random walk starting from a certain chromosome compartment (from a 0.2 to 1 Mb bin) will likely stay within the same compartment, this process can enhance compartmental features (Supplementary Fig. 7a) but lacks the resolution to enhance finer structures that are required to distinguish neuron subpopulations. Finally, random-walk does not significantly affect the embedding of neuron subtypes by deep-learning-based tools (*Higashi* and *Va3DE*) (Supplementary Fig 8d–f).

*SnapATAC2* is a top performer that works well with most benchmark datasets. It performs an explicit feature selection (default 500,000 strongest contacts) before normalization with IDF (inverse document frequency) transformation. IDF improves embedding by giving higher weights to features that are unique to fewer cells. However, in Hi-C matrices distal contacts are sparser and therefore more likely to be unique; conversely, short-range contacts tend to get lower weights because they are less sparse. Therefore, IDF-transformation may lead to a bias towards long-range contacts, which may affect the embedding of neuron subtypes based on the discussion above. We tested this theory by applying IDF transformation prior to *InnerProduct* embedding and found that IDF indeed hurts the embedding of neuron subtypes in human PFC data but does not affect the non-neuron cells (Supplementary Fig. 9a, b). As expected, IDF also does not significantly impact the embedding of mouse oocyte/zygote dataset (Supplementary Fig. 10c, d).

### Deep learning tools require fewer reads to recognize multi-scale genome architecture

There is a growing interest in applying deep-learning-based embedding tools for scHi-C data analysis, but deep-learning methods are hard

to implement. Our benchmark finds that *Higashi* and *Va3DE* are the top two methods that perform well across multiple applications (Fig. 1c–f), suggesting that deep learning tools can recognize different biological priors at both short- and long-ranges. However, since "conventional" tools such as *SnapATAC2* and *InnerProduct* also deliver solid performance with much lower computational burden (Fig. 1d, e), it remains unclear if or when deep learning tools should be applied for single-cell Hi-C embedding. To explore this question, we perform "stress tests" with the embedding tools by decreasing read depth or increasing resolution.

An early study estimated that roughly 5k *cis* contacts per cell is the lower-limit to analyze the cell cycle scHi-C data[22]. Most of the benchmark datasets we used in this study have higher read depth at 50–500 K *cis*-contacts per cell except that the sciHi-C based synthetic mixture datasets have 5–20 K average read depth (Fig. 1a). However, most of the embedding tools still work very well on the synthetic mixture datasets despite the low read depth (Fig. 1c, d), suggesting that the synthetic cell mixtures do not reflect the challenges to embed cell states or populations from natural systems. Therefore, we primarily used the human PFC[5] and human brain atlas[17] scHi-C datasets for the stress tests. For these complex brain tissue datasets, we monitored both "overall ARI" (for all cell types) and "neuron ARI" (for neuron subtypes) because, as discussed in Fig. 5, the embedding of non-neurons and neuron subtypes require long- and short-range contacts, respectively.

Starting from "conventional" embedding tools, when we analyze human PFC dataset with 100% data, *SnapATAC2* obtains slightly better overall ARI than *InnerProduct* (Fig. 6a, quantitation in the left panel of

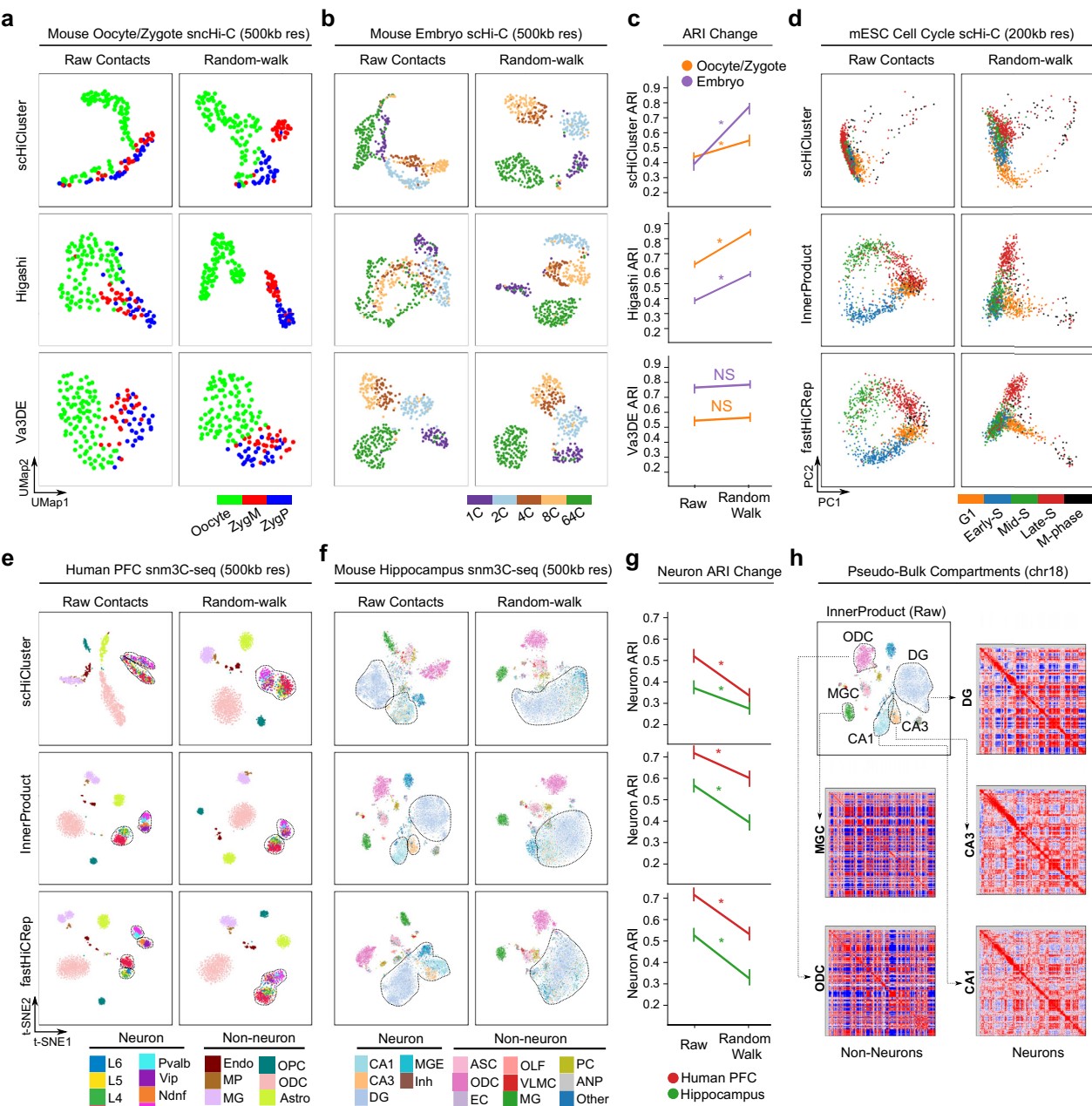

**Fig. 5 | Random-walk is biased towards embedding high-order contacts.**
**a** Embeddings mouse oocyte/zygote data using *scHiCluster, Higashi,* and *Va3DE* with and without random-walk imputation. **b** Compare the tools in (**a**) with the mouse embryo dataset. **c** Summarize the changes of ARI for the analysis in (**a**, **b**). *n* = 5 runs for each method. Data are presented as mean values ± SEM. Two-sided t-test was used to compare raw contact results to random walk results. Oocyte/Zygote: *scHiCluster* t = 13.19 (*p* < .00001), *Higashi* t = 17.9 (*p* < 0.00001), *Va3DE* t = 1.53 (*p* = 0.166). Embryo: *scHiCluster* t = 31 (*p* < 0.00001), *Higashi* t = 18.85 (*p* < 0.00001), *Va3DE* t = 1.66 (*p* = 0.135). Significant *p* < 0.01 is indicated by * and p > 0.01 is indicated by NS. **d** Embeddings the mESC cell cycle dataset using *scHiCluster, InnerProduct,* and *fastHiCRep* with and without random-walk imputation. **e**, **f** Three methods (*scHiCluster, InnerProduct, fastHiCRep*) are evaluated on the human PFC and mouse hippocampus datasets with and without random-walk.

Neuron populations are highlighted with dash circles. **g** ARI for the neuron cell populations from human PFC and mouse hippocampus datasets with and without random-walk. *n* = 5 runs for each method. Data are presented as mean values ± SEM. Two-sided *t*-test was used to compare raw contact results to random walk results. Human PFC Neurons: *scHiCluster* t = 15.64 (*p* < 0.00001), *InnerProduct* t = 9.13 (*p* = 0.000017), *fastHiCRep* t = 21.21 (*p* < 0.00001). Mouse Hippocampus Neurons: *scHiCluster* t = 7.95 (*p* = 0.000045), *InnerProduct* t = 11.51 (*p* < .00001), *fastHiCRep* t = 13.71 (*p* < .00001). Significant *p* < 0.01 is indicated by * and *p* > 0.01 is indicated by NS. **h** Pseudo-bulk compartment analysis of two non-neuron and three neuron cell types in the mouse hippocampus dataset. Note that non-neurons (MGC, ODC) have distinct compartment structure, but neurons (DG, CA1, CA3) are all highly similar. Source data are provided as a Source data file.

Fig. 6c). However, when we examine the neuron cells only, *InnerProduct* outperforms *SnapATAC2* because it shows separation of the L2/3, L4, L5, and L6 subpopulations within excitatory neurons, and the Pvalb, Vip, Ndnf, and Sst subpopulations within the inhibitory neurons (Fig. 6b, quantitation in the right panel of Fig. 6c). When we used 20%

of the data, *InnerProduct* completely loses its ability to distinguish neuron subtypes while *SnapATAC2* can still separate the excitatory and inhibitory neurons into two groups (Fig. 6a–c). *InnerProduct* can still distinguish some non-neuron populations at 10% down-sampling, but eventually loses all embedding power when down-sampled to 2%

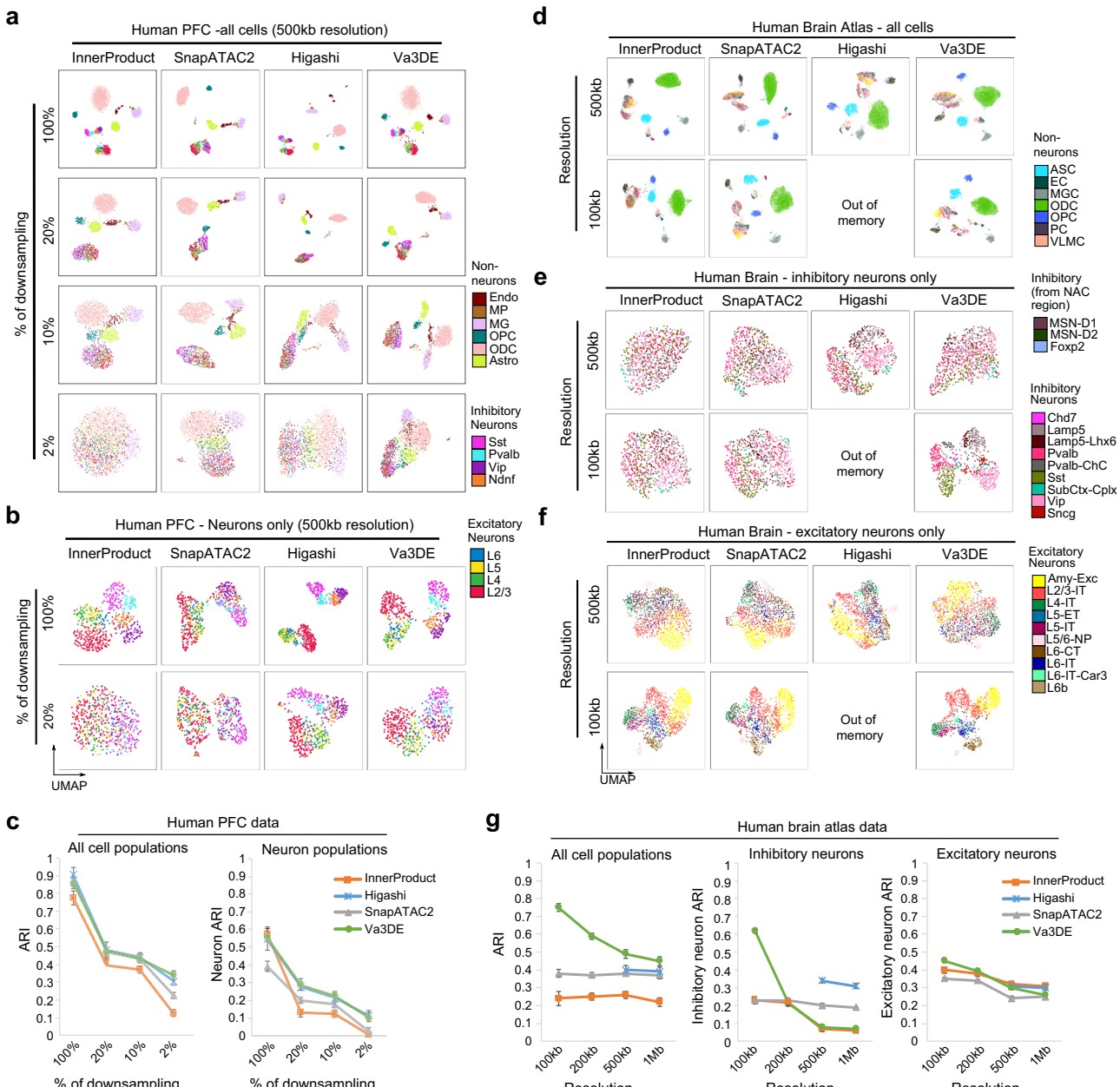

**Fig. 6 | Deep learning methods work better with low-depth or high-resolution data. a** Embeddings of human PFC scHi-C cells using four methods at 500 kb resolution after sampling a fraction of reads for each cell (20%–2%). **b** Embedding visualizations of only the neurons at full read depth and 20% down-sampled depth. Note that *Higashi* and *Va3DE* can still distinguish the inhibitory and excitatory neurons with 20% of the reads. **c** Curves showing how read depth affects the ARI's when using four methods to embed human PFC data. Left: ARIs for all cell types in human PFC are plotted; Right: neuron ARI only for neuron populations are plotted. Note that *Higashi* and *Va3DE* curves are close to each other in these plots. *n* = 5 runs for each method at each downsampled percentage. Data are presented as mean values ± SEM. **d** UMAPs comparing the embeddings of human brain atlas dataset using four methods at 500 kb and 100 kb resolution. *Higashi* could not be run at 200 kb resolution or better. Same as **d** but only the inhibitory neurons **e** or excitatory neurons (**f**) are plotted. **g** ARI vs resolution curves for all cell types (left), inhibitory neurons (middle), or excitatory neurons (right). Results from four resolutions (100 kb -1 Mb) are plotted. Note the substantial improvement of in ARIs for *Va3DE* as resolution increases. *n* = 5 runs for each method at each resolution. Data are presented as mean values ± SEM. Source data are provided as a Source data file.

(Fig. 6a, c). *SnapATAC2* also performs better than *InnerProduct* at the extremes of 10% and 2% down-sampling (Fig. 6a). We believe that *SnapATAC2* works better with low-depth data due to its explicit feature selection step for strong contacts (**Methods**).

The two deep learning-based methods (*Higashi* and *Va3DE*) clearly performed better in stress tests because: (i) with 100% data, both *Higashi* and *Va3DE* can also separate the eight neuron sub-populations like *InnerProduct* (Fig. 6b, quantitation in the right panel of Fig. 6c); (ii) they can also separate non-neuron cells with the extreme

10% and 2% down-sampling like SnapATAC2 (Fig. 6a, quantitation in the left panel of Fig. 6c); (iii) they even retain the capability to distinguish the neuron subtypes with 20% down-sampling (Fig. 6b, c).

To summarize, when read depth gets lower, more random noises will be introduced to the contact matrices. Because the differences between neuron subtypes are much subtler, they are more likely to be masked by the sparsity noises. Deep learning tools better tolerate low read-depth and are also more versatile in recognizing both short- and long-range genome architectural features.

### Deep learning tools better support high resolution embedding

High-resolution embedding is desirable for complex tissue analyses because highly similar cell populations or cell states may only involve subtle changes of local chromatin interactions. However, increasing Hi-C resolution also exacerbates the data sparsity and lowers the embedding performance. To evaluate how resolution affects the performance of different tools, we focus on the human brain atlas data because resolution is particularly important to embed this dataset: out of the 29 total annotated cell populations, there are 10 excitatory neuron subtypes, 9 inhibitory neuron subtypes, and 3 inhibitory interneuron subtypes from NAc region; only 7 populations are non-neurons.

At the standard 500 kb resolution, all the top four tools (*Va3DE*, *Higashi*, *SnapATAC2*, and *InnerProduct*) can distinguish the non-neurons very well (Fig. 6d, Supplementary Fig. 10a). Therefore, the overall ARIs are low mainly because it is difficult to embed the many neuronal subpopulations (Fig. 6g, left panel). Furthermore, the three major neuron groups (excitatory, inhibitory, and inhibitory inter-neurons from NAc) are also distinguishable in the 500 kb-resolution UMAPs (Fig. 6d and Supplementary Fig. 10a, b). We therefore computed the ARI's for the 9 inhibitory (Fig. 6e, g and Supplementary Fig. 10c) and the 10 excitatory neuron subtypes separately (Fig. 6f, g and Supplementary Fig. 10d) as we increase the embedding resolution to 100 kb. Note that *Higashi* failed the test at 200 kb resolution or better due to memory limits (Fig. 1e and Supplementary Fig. 10). In fact, *Va3DE* is the only deep learning tool that works with this dataset at both 200 kb and 100 kb resolutions.

As the resolution increases, we observed that *Va3DE* shows dramatic improvements with both overall ARI and neuron subtype ARIs; the improvement is particularly striking from 200 kb to 100 kb as seen from both the UMAPs and ARI quantitation (Fig. 6e–g and Supplementary Fig. 10), indicating that high resolution embedding can be very helpful to resolve highly similar cell subpopulations. However, increasing resolution does not significantly improve the overall performance of the "conventional" tools (Fig. 6g), In fact, most "conventional" tools achieve best score at 500 kb, not 200 kb resolution (Fig. 1d). For example, as the resolution increased from 1 Mb to 100 kb, both SnapATAC2 and *InnerProduct* show improved performance embedding neuron subtypes, but the overall ARI does not change (Fig. 6g), suggesting that worsening sparsity is negatively impacting the embeddings.

### Multi-modal integration best embeds highly similar cell populations

Embedding highly similar cell populations in complex tissues, such as the neuron subtypes in the brain tissues, remains challenging because it becomes much harder for the embedding tools to recognize specific high-resolution interactions from increasingly bigger and noisier contact matrices. One solution is to develop co-assaying technologies to generate multimodal data, including both scHi-C and DNA methylation or transcription profiles[5,18,19,42,43] (vertical integration[44], see Additional Discussions about co-assay data embedding in **Methods**), but single cell multiome protocols can be hard to implement (e.g., RNA may degrade during sample collection), and it is impossible to add more experimental modality to existing scHi-C data. One alternative approach is to use independent scRNA-seq data to assist the embedding of scHi-C cells (diagonal integration[44]). Diagonal integration is now routinely used to embed scRNA-seq and scATAC-seq data jointly[45].

*ScGAD* has an average rank among all tools when used to embed complex tissue datasets (Fig. 1c). However, *scGAD* was also designed to allow integration with single-cell RNA-seq data. This is possible because *scGAD* aggregates interaction signals around each gene into a single gene score[26], which matches the feature space of the scRNA-seq dataset so that existing integration tools such as *Harmony*[46] can be used to project each scHi-C cell onto a shared embedding space.

Indeed, when utilizing an independent human brain scRNA-seq dataset[47] to perform diagonal integration with *scGAD* gene score matrices, we can achieve a significant increase in the neuron ARI from 0.51 to 0.82, which is higher than any other tools we have tested thus far (Fig. 7b). We therefore believe that multi-modal integration is the most promising approach to embed highly similar cell populations from complex tissues.

## Discussion

It should be noted that although our study primarily compares the embedding tools in clustering analyses, the goal is to seek best options and models to represent the scHi-C data. Since technology biases may impact on the benchmark results, it is plausible that including datasets from more sources would lead to more robust conclusions. Therefore, in this study, we chose ten scHi-C datasets across different protocols with reliable cell type annotations for benchmark analysis. We demonstrated that changing data representation can significantly improve a tool's embedding performance. This conclusion should apply to all downstream scHi-C analyses, including but not limited to cell clustering. We also suggest that, as more scHi-C datasets are collected and new protocols are proposed, some conclusions of this benchmark should be further tested to find the optimal data representations for more scHi-C applications.

We performed a comprehensive evaluation of multiple single-cell Hi-C embedding tools in different applications and offers insights into why some methods are better than others depending on biological contexts. The results also elucidate the dominant variations between different cell states or cell populations at single cell level. We found that long-range interactions mark various stages of early embryogenesis (Fig. 2) but short-range (<2 Mb) are necessary to characterize the more similar cell populations in complex tissues (Fig. 3). Regarding cell cycle, while mitotic cells require long-distance interactions for embedding, the G1, early-, mid- and late-S phases can be only distinguished with short-range contacts (Fig. 4). Equipped with knowledge about biological priors, it is possible to improve both conventional and deep learning pipelines by choosing the right settings or preprocessing steps, such as a random walk, IDF, and distance truncation.

Clustering from single cell embedding is the most common application of scHi-C analysis. After clustering, imputation tools are now available to extract cell population- or cell state-specific architectural features at all scales from pseudo-bulk data[21,27,48–52]. Most of the embedding tools work very well at low resolution (500 kb) in clustering cells from different lineages, such as the synthetically mixed cell lines and the non-neuron cell types in complex brain tissues. Distinguishing cell populations from the same lineage, such as the neuron subtypes, is more challenging and sometimes need high resolution embedding at 200 kb or better. We also showed that preprocessing operations need to be performed with caution. Specifically, random-walks in *scHiCluster* emphasize compartment-scale structure but smooth out short-range heterogeneity (Fig. 5, Supplementary Fig. 8); IDF transformation in *SnapATAC2* is also distance biased because long-range contacts get more weight with increased data sparsity (Supplementary Fig. 9). These insights will help users to make informed decisions in choosing the right tools or settings when embedding new scHi-C datasets.

The major advantages of deep learning pipelines are the versality with multi-scale architecture variations in different applications, and the robustness with low read-depths. In fact, both top two methods in our benchmark are deep learning tools (*Higashi* and *Va3DE*). Therefore, we recommend deep learning tools to obtain best results if cell number is not too high (e.g., <10 K cells), starting from 500 kb resolution, unless the research wants to focus on highly similar cell subpopulations. Deep learning methods are also preferred when the biological prior is unclear or when the average read depth is low.

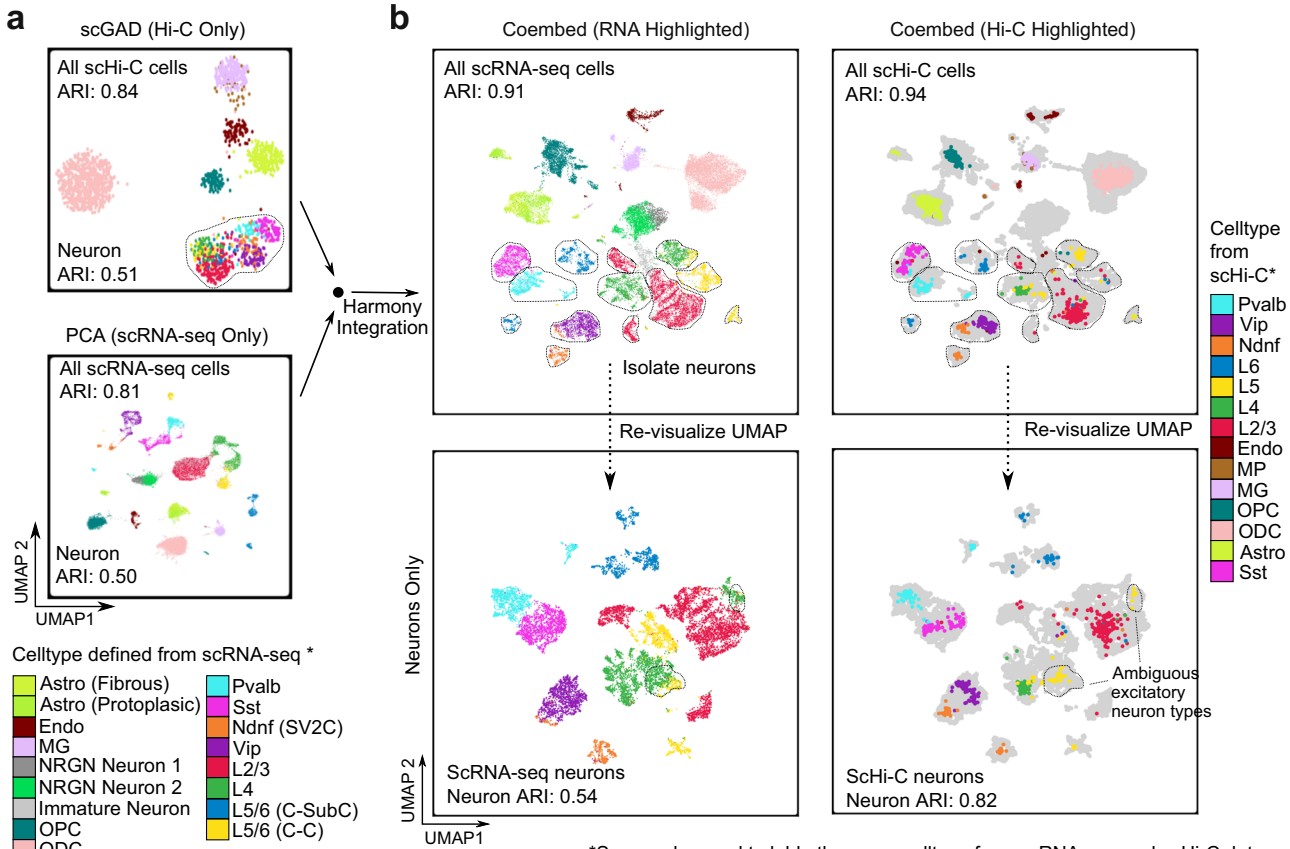

**Fig. 7 | Integration with scRNA-seq data improves scHi-C embedding performance. a** Standalone embeddings human PFC scHi-C data with *scGAD* (up panel) and an independent human brain scRNA-seq data (bottom panel). **b** Integration of scHi-C with scRNA-seq embedding using *Harmony*. Top left: scRNA-seq cells visualized in joint embedding space; Top right: scHi-C cells visualized in co-embedding space (scRNA-seq data is grayed out); bottom left: only neurons from scRNA-seq data are visualized in co-embedding space; bottom right: only neurons cells from scHi-C data are visualized in co-embedding space. Note that nearly all neuron subtypes are distinguishable in the joint embedding.

However, when cell number is high, it will be beneficial to use conventional methods for high efficiency. Particularly, we recommend *SnapATAC2* (at 500 kb resolution) as it ranks the highest among all conventional tools and shows robust performance across all the benchmark datasets. *SnapATAC2* also works well with low-depth data due to its explicit feature selection. However, *SnapATAC2* performs less well with neuron subtypes even at high resolution (Fig. 6) due to the bias of IDF-transformation toward distal contacts (Supplementary Fig. 9). *InnerProduct* is an alternative conventional tool that performs better at high resolution to detect short-range heterogeneity, but the performance of *InnerProduct* is vulnerable to low read depth (Fig. 6).

We recommend *Higashi* or *Va3DE* for high resolution embedding (200 kb or better) to resolve subtle differences between cell sub-populations. We note that *Higashi* can also explicitly impute contacts for single cells, which can be useful for single cell heterogeneity analysis (see Additional Discussions in **Methods**). However, the memory limits of *Higashi* prevent its high-resolution applications when cell number is high. *Va3DE* is less memory demanding because single cell data can be loaded in batches. We pushed *Va3DE* to 100 kb resolution and achieved the best embedding of neuron subtypes in the largest human brain atlas dataset (Fig. 6d–g). We noticed that the runtime of *Va3DE* increases proportionally to the cell number while *Higashi* has near constant (but longer) runtime as cell number increases (Supplementary Fig. 11, Additional Discussions in **Methods**). Further work is necessary to improve the memory and time efficiency of deep embedding methods at high resolution. Finally, clustering highly similar cell populations or states may not be achievable with scHi-C

data alone. Encouraged by *scGAD*, we believe that multiome "diagonal integration" using independent scHi-C and scRNA-seq data will be a very useful direction.

## Methods
### Embedding method details
**1D-PCA**. A simple baseline method that attempts to represent scHi-C data like typical 1D single-cell data. Ignoring the 2D interaction information, we simply count the number of interactions at each bin to generate a genome-wide bin count vector for each cell. The cell X bin matrix is then represented using PCA. This is in contrast to "2D" baseline, which unravels each 2D contact matrix into a larger 1D vector. This is what *scHiCluster* does and can be considered an analogous "2D-PCA" baseline to this 1D-PCA baseline. However, the preprocessing pipeline prior to unraveling the chromosome contact matrices is what makes scHiCluster unique (see below).

**ScHiCluster**. *ScHiCluster*[21] embeds each cell by first normalizing each chromosome contact matrix (VC_SQRT_NORM), then imputing interactions by applying a box filter followed by a random walk. After filtering the top 20% strongest interactions, PCA is performed separately for each chromosome, followed by a full genome PCA step on the concatenated chromosome PCs. Performing *scHiCluster* with no preprocessing steps is analogous to a "2D-PCA" compared to our 1D-PCA baseline. We utilize the scHiCluster implementation included in the scHiCTools package (https://github.com/liu-bioinfo-lab/scHiCTools/).

**InnerProduct and fastHiCRep.** *InnerProduct*[23] is a stratified method that embeds each cell using multi-dimensional scaling (MDS). To compute the distance matrix for MDS, *InnerProduct* considers each stratum of the contact matrices and computes the inner product between the z-normalized strata vectors and averages the similarities of each stratum to produce a single similarity value for each chromosome for each pair of cells. The chromosome similarity matrices are then aggregated together using either mean or median to produce the final similarity values that MDS will attempt to preserve. *fastHiCRep*[23,38] is a special case of *InnerProduct* where distal strata carry a higher weight than short-range strata. This is the stratum-adjusted correlation coefficient introduced by Yang et al. [38] We run these methods using the scHiCTools package (https://github.com/liu-bioinfo-lab/scHiCTools/).

**cisTopic.** *CisTopic* was originally developed by Gonzalez-Blas *et al.* for scATAC-seq analysis[53] and later adapted by Kim et al. for scHi-C data. This method translates the problem of cell representation learning into a problem of topic modeling, a common method used in text analysis and document summarization via the bag-of-words model. In our case, a "word" is an interaction between two genomic loci, and a "document" is a cell. The method first looks through every contact found in the dataset across all cells (within a preset maximum range) to create a list of all the possible locus pairs. The dataset is then represented as a cell-locus pair occurrence matrix, and we can apply LDA to learn a set of latent topics that best explain the observed locus pairs across the dataset. We use the provided scripts from (https://github.com/khj3017/schic-topic-model/)[20] and *cisTopic* R package (https://github.com/aertslab/cisTopic)[53]. We also provide a simple Python implementation of the *cisTopic* method in our *SCORE* package to avoid having to install extra dependencies.

**SnapATAC2.** While initially designed for scATAC-seq embedding, it shows promise in embedding scHi-C data as well[24]. It uses an identical data representation as cisTopic building up a locus-pair set. SnapATAC2 filters out interactions based on an upper and lower quantile (0.5% for both by default) of total counts in the entire dataset and then keeps the 500k interactions with strongest average signal. It then performs inverse document frequency (IDF) transformation followed by spectral decomposition of the pairwise cell similarity matrix (using cosine distance). This contrasts with the original SnapATAC[54], which used a binary representation of each feature and measured pairwise Jaccard index to generate the pairwise cell similarity matrix. We utilize version 2.7.0 of SnapATAC2 (https://github.com/kaizhang/SnapATAC2).

**InsScore.** *Insulation Score* is a measure introduced by Crane et al. [30] to model low resolution TAD boundaries in bulk Hi-C. By aggregating the total interactions within a sliding window, this method computes a delta vector along the whole contact matrix. When this delta vector crosses zero, we interpret that as a domain boundary. To embed single cells, we use this delta vector directly as a cell representation and embed the dataset using PCA. We use the Perl implementation provided by the authors (https://github.com/dekkerlab/crane-nature-2015/)[30]. We also provide a simple Python implementation in our *SCORE* package to avoid having to install extra dependencies.

**deTOKI.** *deTOKI* identifies TADs using a similar sliding window technique, but with an explicit goal of interaction clustering. Each submatrix in the sliding window is decomposed using repeated non-negative matrix factorizations (NMF)[55]. Using the decomposed submatrix, *deTOKI* clusters bins together into TADs and identifies the optimal number of TADs using the Silhouette coefficient[56]. To embed cells using the predicted TAD boundaries, we summarize the reads density in each TAD for each cell and use this representation to embed

the dataset. We use the Python implementation provided by the authors (https://github.com/lixiaoms/TOKI)[31].

**scGAD.** *scGAD*[26] is a technique developed for integrative analysis of scHi-C with other single-cell data modalities. The goal of this method is to transform the scHi-C data into a representation that can be projected onto other single-cell data, such as scRNA-seq. A typical representation of scRNA-seq data is in the form of a cell-gene matrix. Decomposing this matrix is a key step in scRNA-seq dimension reduction. By transforming a scHi-C dataset into a cell-gene matrix, we can produce a joint embedding of the two data modalities using cross-decomposition methods such as CCA. The main challenge here is that for a single gene locus, we need to map all interactions to a single value. *scGAD* accomplishes this by first normalizing the data using their *BandNorm*[28] method, then computing z-scores at each gene locus. This produces a gene-score for each locus across each cell. We can embed this matrix directly as in our comparisons, or we can project onto other single-cell datasets. We run *scGAD* using version 0.1.0 of the BandNorm package (https://sshen82.github.io/BandNorm/). To co-embed scHi-C with scRNA-seq data using *scGAD*, we utilize the *Harmony*[46] package implemented in *scanpy* (https://scanpy.readthedocs.io/en/stable/generated/scanpy.external.pp.harmony_integrate.html).

**scVI-3D.** *scVI-3D* is a deep generative model that also relies on the *BandNorm*[28] normalizing technique. This method considers each band (strata) of the contact matrices independently, training a variational autoencoder (VAE)[57] to represent each strata across the whole dataset. These strata representations are concatenated together to form the full embedding of each cell. We use the implementation provided by the authors (https://github.com/yezhengSTAT/scVI-3D), which implements the VAEs using the scVI-tools Python package (https://scvi-tools.org/).

**Higashi.** *Higashi*[27] is unique among all methods considered here as it is the only one to use a hypergraph representation. The whole dataset is represented as a massive set of (cell, bin, bin) hyperedges, and the network is trained to predict the probability of a given (cell, bin, bin) triplet being present in the dataset. *Higashi* is a manifestation of recent developments in geometric deep learning[58] and graph neural networks[59,60]. While convolutional neural network (CNN) models such as *Va3DE* train on the neighborhood structure of individual strata or large stacked matrices, this is not necessarily the most effective and accurate geometric prior. For example, a single bin $b_1$ may form a short-range interaction with $b_2$ and a long-range interaction with $b_3$. In CNN models, $b_2$ and $b_3$ are neighbors via a very distal path of pixels; a relationship that a single convolution may miss. *Higashi* trains directly on the geometric structure of the data, and so $b_2$ and $b_3$ are second-order neighbors in the graph, and the training process can easily capture this relationship via a graph convolution. The hypergraph data representation also allows *Higashi* to have additional functions, including full hypergraph data imputation and integration with other modalities. The quality of embedding also impacts the performance of data imputation and multi-modal integration, but this is out the scope of this study. We utilize the preprocessing and training scripts provided with the original publication (https://github.com/ma-compbio/Higashi).

**Fast-Higashi.** *Fast-Higashi*[25] constructs a (cell, bin, bin) hypergraph in the same way as Higashi, but instead of training a neural network to embed nodes and solve the hyperedge prediction problem, Fast-Higashi decomposes the full cell X bin X bin tensor using the PARAFAC procedure. We utilize the base Higashi preprocessing scripts (https://github.com/ma-compbio/Higashi) and then run the Fast-Higashi training procedure (https://github.com/ma-compbio/Fast-Higashi).

**Va3DE.** Variational Deep Embedding (*VaDE*)[32] is a generative model that assumes the data are generated according to latent samples drawn

from a Gaussian Mixture manifold (Supplementary Fig. 12). It does this by replacing the single standard *Gaussian* prior of a variational auto-encoder (VAE)[57] with a *Gaussian* mixture prior. Thus, the encoder network learns to embed each datapoint in a low-dimensional space according to a learned cluster assignment. We assume that the data are generated according to the joint probability:

$$p(\pmb{x}, \pmb{z}, c) = p(\pmb{x}|\pmb{z})p(\pmb{z}|c)p(c) \qquad (1)$$

where *x*, *z*, and *c* are random variables representing a data sample, encoded representation, and cluster assignments, respectively. *VaDE* maximizes the data-likelihood $p(x)$:

$$\log p(\pmb{x}) = \log \int_{\pmb{z}} p(\pmb{x}, \pmb{z}, c)d\pmb{z} \qquad (2)$$

We assume that $p(c)$ is a categorical distribution, $p(\mathbf{z}|c)$ is a cluster-specific Gaussian, and $p(\mathbf{x}|\mathbf{z})$ is either a *Poisson* distribution representing the counts of each interaction or a *Bernoulli* distribution representing the presence of an interactions at any point of the contact matrix. For all benchmark results, we use the default Poisson distribution, however, for highly sparse datasets, a binary representation may be more stable. *VaDE* has successfully been applied to single-cell ATAC-seq analysis[61,62] where each cell is a 1D vector and both $p(\mathbf{z}|c)$ and $p(\mathbf{x}|\mathbf{z})$ can be approximated by fully-connected neural networks. However, in the case of scHi-C data, we need to account for the unique 2D structure of contact matrices. To accomplish this, we replace the fully connected encoder/decoder of *VaDE* with a convolutional encoder/decoder to help capture the spatial structure of each contact matrix. The new method is named *Va3DE*.

We use 4 convolutional blocks, each with two Conv2D layers, with the last layer of each block having stride 2 in at least one dimension to reduce dimensionality at each block. The final feature maps are flattened and mapped to the latent distribution using a single fully connected layer. The sampled latent vector is mapped back to the input space using 4 convolutional blocks, each with two Conv2DTranspose layers, with the last layer of each block similarly increasing dimensionality as in the encoder. The final feature maps are aggregated together using a 1 × 1 convolution to map to the output distribution parameters (Poisson or Bernoulli mean). We use an increasing number of filters in the encoder (4, 8, 16, 32) and a decreasing number of filters in the decoder (32, 16, 8, 4). The model minimizes the negative log-likelihood of the output distribution with a KL-divergence regularization term for the learned GMM mixture prior. We use the *Adam*[63] optimizer with a learning rate of 3e-4 and weight decay of 1e-4 by default. We find that most embeddings stop improving after ~1000 epochs, though small datasets can take more. Empirically, we find that the model should be trained on ~2 M cell observations (e.g., 2000 cells trained for 1000 epochs).

Time and memory constraints typically prevent us from training a model on the full contact matrices of each cell, so we extract a fixed number of strata from each contact matrix and stack them into a wide genome band matrix. For this reason, *Va3DE* can technically only be used in a limited distance context, as it is unable to use all the reads in a cell. This is not an issue for datasets that require long-range interactions since we can typically represent these datasets at low resolution. For datasets where we must consider long-range interactions, and memory constraints are still a barrier to training the model, we can extend the limited range of *Va3DE* by ignoring short/mid-range interactions. *Va3DE* and *scVI-3D* are similar in that they both train generative neural networks; however, *scVI-3D* considers each stratum independently. This simple extension allows *VaDE* to consider the global contact matrix structure rather than each individual distance. Thus, we sacrifice more computational time training but for higher clustering accuracy in many scenarios.

An important hyperparameter in building a *Va3DE* model is the number of Gaussians to model in the mixture (i.e., the number of total clusters to model). In theory, this should be the exact number of cell types in our dataset, however, we often do not have this information ahead of time. We explored the dependence of ARI on this hyperparameter and found that highest clustering accuracy is achieved when we *overparameterize* the model (i.e., select more clusters than necessary) (Supplementary Fig. 13). This is because it allows *Va3DE* to model intra-cell type heterogeneity and assign multiple gaussians to a single cell type. Unused gaussians simply have their prior probability pushed to zero as the model converges.

We implement *Va3DE* in TensorFlow Probability version 0.11.1 and TensorFlow version 2.3.1.

## Method memory complexity details

**1D-PCA**. Runs PCA on a $N X M$ matrix so memory bottleneck is the same as PCA: O*(min(N, M) \* M)*.

**scHiCluster**. Runs PCA on full chromosome contact matrices (unraveled/flattened), which are $NxM^2$ matrices but stored in sparse format. However, the memory bottleneck comes from PCA step during which dense matrices are created: O*(min(N, M²) \* M²)*.

**InnerProduct and fastHiCRep**. Must construct a pairwise similarity matrix to compute MDS, so memory usage is quadratic with respect to the number of cells O*(N²)*, and we need to be able to store each of the $k$ strata O*(Mk)*, making total memory usage O*(N² + Mk)*.

**cisTopic**. We only need to store the full set of locus pairs for each cell, and then the model can be trained efficiently using Gibbs sampling. Using $k$ strata, we end up having to store a matrix of size O*(NMk)*.

**SnapATAC2**. Uses an identical data representation as cisTopic to store a matrix of size O(*NMk*).

**Fast-Higashi**. Stores data matrices in sparse format. But during initiation, it does random-walk by default and attempts to generate a full $N X M X M$ hypergraph prior to training O*(NM²)*. Note that random-walk option can be disabled to improve memory usage, but the authors recommend disabling random-walk only when the data quality is high. In this study, we keep the random-walk option on for all benchmark analyses. We noticed that under default conditions, *fast-Higashi* is more memory demanding than *Higashi* because *fast-Higashi* failed 200 kb resolution with a 1.8 K cell dataset but *Higashi* still works.

**InsScore and deTOKI**. The sliding window used to compute insulation scores and deTOKI TAD boundaries does so while only ever needing to store a fixed number of strata in memory. Then we need to be able to store the full insulation score vectors for each cell, making the total memory usage O*(NM + k)*.

**scGAD**. For each cell, we generate a small contact matrix around the gene body. Because of this, scGAD technically is quadratic with respect to resolution but with a much smaller constant than any other method, since each contact matrix is bound by the size of the largest gene. We do this for each cell, making the total memory requirements O*(NM²)*.

**scVI-3D**. The same as cisTopic, we must construct the full band matrix for each cell to perform the *BandNorm* normalization step required making the memory usage O*(NMk)*.

**Higashi**. The main memory bottleneck of Higashi is the preprocessing rather than training. We must construct the entire $N X M X M$ hypergraph prior to training which requires O*(NM²)*.

**Va3DE.** During training we randomly load batches of cells at a time where we must store a constant number of strata. This step requires *O(Mk)*. After training, we can predict embedding coordinates on batches of cells the same as training, completely avoiding ever loading all cells in memory except as embedding coordinates.

### Hardware details

Experiments were run on a system with 256GB memory, 24 cores, and a NVIDIA A30 GPU (24GB VRAM). This was sufficient for all methods requiring GPU (*Higashi, Fast-Higashi, Va3DE*) except *scVI-3D* at high resolutions and cell counts (e.g., human brain at 200 kb resolutions or better) when it attempted to load the entire dataset band matrix into GPU memory. We additionally note that all methods can be run on a NVIDIA GeForce GTX 1080Ti (11GB VRAM), which is an old consumer GPU. Overall, GPU memory usage is not an issue for any of the benchmark tests. The main bottleneck we find is in RAM usage during preprocessing, which limits most methods to lower resolutions.

### Dataset QC

We adopt the same cell filtering strategy as described by *scHiCluster*. Any cell with fewer than 5k *cis*-contacts is automatically filtered. We then scan each chromosome and confirm that all chromosomes of length *x*-Mb have at least *x* contacts in that chromosome. If a cell has a single chromosome that does not satisfy this, we filter the cell. Some dataset-specific QC strategies are also done as described below.

**Mouse embryo.** For all benchmarking analysis, but not for the cell-cycle analysis in Fig. 4, we filter the mitotic cells as described by Collombet et al. We plot the short-range contacts (25 kb–2 Mb) vs long-range contacts (2–12 Mb) and fit an ellipsoid to the data. Using a cutoff point of (0.15, 0.35) and the center of the ellipsoid, we include only G-S phases as all data points in the left ascending side of the ellipse. Other cells are assumed to be mitotic and filtered.

**mESC cell cycle (Nagano).** We additionally filter cells with fewer than 50 k *cis*-contacts and drop the mitotic cells when performing benchmarking.

**Human prefrontal cortex.** We additionally filter cells with fewer than 50 k *cis*-contacts.

**Mouse hippocampus.** We additionally filter cells with fewer than 50 k *cis*-contacts.

**Human brain atlas.** We additionally filter cells with fewer than 100 k *cis*-contacts.

### Method evaluation details

We run each method using the recommended parameters, except in cases where we explicitly vary the settings. Each method outputs a low-dimensional representation of each cell in the dataset. We primarily used K-means in the benchmark for clustering (see **Additional discussion** below about the choice of clustering methods.

Six embedding tools (*1D-PCA, fastHiCRep, InnerProduct, InsScore, scGAD, SnapATAC2*) are deterministic (generate identical embedding in all repeats), and the other methods are not. The deterministic embedding methods can still vary in clustering performance due to the stochastic clustering algorithms. We therefore repeat each embedding at least 5 times to get a distribution of performance while balancing the significant runtime of some of these analyses.

We utilize three main metrics to evaluate the performance of each embedding method: Adjusted Rand-Index (ARI), Normalized Mutual Information (NMI), and cell type Average Silhouette Scores (ASW). ARI takes in a predicted clustering assignment and can be interpreted as an

accuracy score measuring if cells from the same population belong to the same cluster, with a correction for cluster sizes. NMI is an independent metric from ARI, which also measures the quality of a clustering assignment compared to the known cell types. ASW measures the concordance of the embedding itself with known cell types and is independent of the clustering algorithm.

ARI can be considered as a chance-corrected form of the Rand Index (RI), which is a metric between 0 and 1 that converts the clustering accuracy computation into a binary accuracy computation[33]. Consider two cells, then define a true positive (TP) classification as same ground-truth cell type and same cluster assignment, true negative (TN) as different cell type and different cluster, false positive (FP) as different cell types but same cluster, and false negatives (FN) as same cell type but different clusters. Rand Index is defined as the binary accuracy across all possible pairs of cells: RI = (TP + TN)/(TP + FP + FN + TN). The ARI is then adjusted for chance (due to, e.g., variable sizes of clusters and cell types) based on: (RI - Expected_RI) / (max_RI - Expected_RI).

NMI is computed based on mutual information, which is then normalized to the range 0 to 1. Mutual information is defined based on the ground truth labels (*U*) and the cluster assignments (*V*) as:

$$MI(U, V) = \sum_{i=1}^{|U|} \sum_{j=1}^{|V|} \frac{\left| U_i \cap V_j \right|}{N} \log \frac{N \left| U_i \cap V_j \right|}{\left| U_i \right| \left| V_j \right|} \tag{3}$$

Average Silhouette Scores are computed as the mean of all sample silhouette coefficients. For each cell, we define the mean intra-cell type distance in the embedding and the mean distance to the nearest cell types. The silhouette coefficient of this cell is the difference between these distances, normalized between −1 and 1. We use cosine distance to measure silhouette coefficients.

Although we primarily use ARI to quantify the clustering performance, we also report the normalized mutual information (NMI) and cell type average silhouette scores (ASW) (see source data for Supplementary Fig. 2). We also use the AvgBIO score, which is an average of the ARI, NMI, and ASW, to summarize the clustering results.

### Per-strata inner product heatmaps

To better interpret the embedding behavior of baseline methods, we run the *InnerProduct* pipeline but stop before performing any aggregation. We compute all the per-chromosome and per-strata correlation matrices and first take the median across each chromosome, as is typical of the embedding pipeline. However, instead of producing one final similarity matrix for MDS embedding, we compute the median similarity at each stratum for each cell. So, if a certain stratum contains highly correlated interactions among a group of cells, these similarities will be highlighted. Though since we are still just operating on a standardized transformation of the raw counts, we see similar distance decay as in raw contact matrices, so the distal correlations tend to be much lower than shorter-range values. For this reason, each heatmap is normalized for each strata column, where 0 indicates regions of dissimilarity and 1 indicates regions of high similarity. While we lose some information by needing to take a median along one of the cell dimensions, this technique allows us to better interpret which distances are being used to embed cells when we run MDS.

### Force-directed graph visualization of chromosomes with pseudo-bulk scHi-C data

To visualize the structure of an individual chromosome, we pool together the raw reads from all cells of a population to produce a pseudo-bulk contact matrix. We construct a network of bin/anchor nodes and interaction edges using *NetworkX*. This can be accomplished using the "score merge" function in our SCORE package. We then use *Gephi* to visualize the chromosome network using the built-in

Force Atlas 2 algorithm with attractive forces proportional to normalized (VC_SQRT_NORM) interaction counts.

## Pseudo-bulk compartment analysis for the mouse hippocampus data

We downloaded the raw pairs files from GSE156683 and merged pairs from the same cell type. The *cis*-reads, which two reads are in the same chromosome, were mapped to the DpnII fragments. The read pairs satisfying one of the following criteria were removed: (1) one of the reads spanning the enzyme cutting site locus (2) two reads were assigned into the same fragment. After filtering, the remaining cis-reads represents two distinct DpnII fragments. According to the strand orientations of both reads, the read pairs were divided into three categories: same-strand, inward, or outward. The "inward" and "outward" read pairs were further filtered, removing those where the distance between two reads exceeds 1 kb and 5 kb, respectively. Finally, the filtered "inward" and "outward" read pairs, along with all "same strand" read pairs, were merged as the cis reads pair, which were assigned into 5 kb anchors to do the compartment calling.

The compartment level analysis is performed following the method described previously in ref. 4. The whole genome was divided into 200 kb bins, and for every chromosome, the contact matrix between bins was generated. The correlation coefficient for two bins, which indicates the similarity of two bins' interaction pattern, was then calculated after normalizing contact matrix by genome distance. All the correlation coefficients constitute a correlation matrix. After the principal component analysis of the correlation matrix, each bin has a PC1 value, and the genome is then divided into two compartments based on whether a bin's PC1 value is positive or negative. Compartments A and B were identified using the H3K4me3 data, where compartment A has more H3K4me3 peaks, while compartment B has less peaks.

## Additional discussion

**Choice of clustering methods for benchmarking.** Our benchmarking SCORE package includes the options of multiple clustering methods, including K-means, GMM (Gaussian mixture model), agglomerative hierarchical clustering, Louvain, and Leiden. Empirically, we note that consistent performance across all clustering algorithms is a good indicator of stable embedding, but since every tool produces embeddings with different geometric properties, various clustering algorithms may perform differently. For example, *Va3DE* is regularized to produce Gaussian-like clusters, and so we would expect a Gaussian Mixture Model (GMM) to fit the embedding well; *fastHiCRep* and *InnerProduct* use MDS to preserve a distance matrix, so it might make sense to cluster them hierarchically by distance.

We performed a systematic analysis comparing the performance of K-means, GMM, agglomerative hierarchical clustering, Louvain, and Leiden for each embedding (Supplementary Fig. 14). Notably, both Louvain and Leiden infer the number of clusters, i.e., they may return different numbers of clusters for different embedding methods, making it difficult to compare their performance. To solve this problem, we vary their "resolution" parameter (referring to the resolution of the clustering algorithms, not the Hi-C data) until the predicted number of clusters matches the expected number. Note this approach is often used in benchmark analysis, including in the recent *SnapATAC2* paper[24]. We found no clear winner among the different clustering algorithms. For example, Louvain and Leiden show slight preferences for human PFC data but perform weak with mouse hippocampus data. Overall, there were no big differences when considering all datasets together (Supplementary Fig. 14). K-means appears to be one of the methods with "stable" performance across different datasets as it rarely becomes the best or worst clustering method with any of the benchmark dataset (Supplementary Fig. 14), suggesting no biases towards or against certain dataset. We primarily used K-means in the benchmark also because most of the tools we benchmarked used

K-means in their original papers. With K-means, it is easier to make all tools output the same number of clusters for fair comparison with ARI and NMI metrics.

**Method runtime details.** We benchmark the runtime of all methods across resolutions and cell numbers to plot how each methods scale with these each parameter (Supplementary Fig. 11). We had to use datasets with a small number of cells for this analysis because some tools will fail due to memory limit when cell number is high. *Higashi* and *deTOKI* take the longest to run as resolution increases, making resolutions beyond 500 kb usually infeasible. *InnerProduct* and *scGAD* prove to be some of the most efficient methods when scaling to high resolution. When scaling the number of cells in the dataset, we find that most methods scale linearly with the number of cells. However, due to the fixed number of training epochs recommended by *Higashi* and the small number of cell nodes compared to bin nodes in the hypergraph, *Higashi* runtime remains relatively constant as we scale up the number of cells, making it attractive for large datasets if its memory and high-resolution efficiency can be improved.

**The difficulty of embedding HiRES and CARE-seq data.** We are surprised to find that the scHi-C components of HiRES and CARE-seq are much harder to embed and cluster on their own. Without using the co-assayed scRNA-seq data, the ARI and AvgBIO scores with HiRES and CARE-seq datasets are much lower than other benchmark datasets, regardless of the tools used (Fig. 1 and Supplementary Figs. 2–3). We noticed that these datasets only have <400 cells. However, when we down-sample the human PFC datasets (to 398 cells) and the mESC cell cycle dataset (to 384 cells), both InnerProduct and SnapATAC2 still achieve much better clustering (Supplementary Fig. 15). We therefore suspect that the changes of protocol in HiRES and CARE-seq (to allow RNA co-assay) may have negatively impacted the quality of scHi-C contacts; this hypothesis still needs further investigation.

**Data imputation in scHi-C analysis tools.** Data imputation is a useful function for certain scHi-C applications. There are two types of scHi-C data imputation: at pseudo-bulk level or at single cell levels. Although our study did not directly benchmark the data imputation functions, here we provide some additional discussion regarding the choice of tools for scHi-C imputation.

1. The quality of pseudo-bulk level imputation is determined by the quality of embedding.

    Most of the tools we benchmarked do not include an imputation function, but if a method can do good embedding and cell type clustering, pseudo-bulk data can be generated based on the clustering and used for imputation with third-party packages. There are multiple high-performance tools to impute sparse Hi-C data at compartment, TAD, or loop levels[50,64–68]. Imputation from pseudo-bulk data is better than imputation from single cells because pseudo-bulk data is less sparse and allows higher resolution. In this scenario, the quality of data imputation is determined by the quality of embedding.

2. Imputation at single cell level is possible but lacks ground truth for benchmarking.

    Three tools (*scHiCluster*, *fast-Higashi*, and *Higashi*) can provide imputed data at single cell levels. The imputation of scHiCluster is "random-walk". In this method, the contacts in every cell form a graph, and the imputation happens for every cell independently before embedding[21]. We concluded from the benchmark analyses that random-walk pays more attention to long-range contacts than short-range contacts, suggesting the limitation of imputed single-cell data matrices.

    *Fast-Higashi* imputes contacts based on a "partial" random-walk which do random-walk before seeing the full contact matrices of the cells. This is an implementation option to save

memory usage, and the performance is nearly identical to the random walk in scHiCluster[25].

*Higashi* uses a unique hypergraph data representation to allow borrowing information from $k$ (default 4) nearest neighboring cells for data imputation after embedding is done. Therefore, this approach can be considered as "local pseudo-bulk imputation" except that there is no need to explicitly pooling the data from neighboring cells. Even without using neighboring cells ($k = 0$), data imputation still benefits from the embedded hypergraph representing all cells in the scHi-C datasets. In their original paper, the authors of *Higashi* already compared *Higashi* imputation to *scHiCluster* imputation and reported that the heatmaps from *Higashi* imputed single cells show common patterns if they are close in the embedding space, and the imputed data also agrees better with a "pseudo-bulk" 3D genome image data[27]. These results all indicate that because *Higashi* imputes the contacts after the embedding, the imputed contacts highlight common 3D genome features between nearby cells in the embedding space, leading to results more resemble the pseudo-bulk contact maps. The random-walk in scHiCluster lacks this ability because it imputes data before embedding.

As explained above, the *Higashi* imputation is still a prediction of 3D genome feature at pseudo-bulk level. However, if the research goal is to study the single cell 3D genome heterogeneity between cells from the same cell population. It is unclear whether *Higashi*'s strategy (imputing after embedding) is the best option, as *Higashi* may suppress cell-to-cell variations within the same cell population. Unfortunately, we cannot benchmark the imputation of cell-to-cell variations because we do not have proper ground truth.

3. Variational autoencoders are not suitable to imputing single cell heterogeneity.

Both scVI-3D and Va3DE algorithm can theoretically also output imputed single cell data matrices from their decoder layers. However, it is important to understand that these tools use variational autoencoders, which eventually represent data as Gaussian distributions instead of fixed values in the latent space and Poisson distributions in the data space. The advantage of variational autoencoders is to obtain better data representation by modeling random noises in the data, thus reducing the risk of overfitting. However, these models by design are not suitable to reconstruct imputed data for single cells because when we input the data of any single cell into these models, they can output infinite numbers of possible imputations due to the Gaussian distributions in the latent space and the data space, so typically reconstructions from variational autoencoders are computed from the mean of these distributions. It is known from the literature that the reconstructed data from variational autoencoders can suffer from blurring and "over smoothing", especially as the latent representations become better and more disentangled[69]. Due to this reason, in Va3DE we only use the reconstruction signal to improve the representations but toss the decoder entirely due to the uncertainty of imputed data. It is not a concern for common applications because, again, it is always possible to obtain better imputation from pseudo-bulk data once good embedding and clustering are achieved.

To summarize, since the goal of most single-cell studies is to analyze cell populations in complex tissues or cell mixes, good embedding and clustering are more important in this scenario because imputation from pseudo-bulk data can be done with third party packages with better accuracy[70]. Single cell imputation is possible with *scHiCluster* (before embedding) and *Higashi* (after or during the embedding). Previous work has already shown that *Higashi* imputation better reflects the common 3D genome features within a cell population at pseudo-bulk level. However, it is not feasible to directly benchmark the imputed single cell heterogeneity due to the lack of ground truth data. The Va3DE and scVI-3D models are designed for reliable embedding, but not for stable imputations.

**Tools that allow co-embedding of co-assay data.** In scHi-C co-assay protocols, the co-assayed scRNA or scMethyl-seq modality can solve the clustering issue regardless of the scHi-C data quality or the choice of scHi-C embedding method. However, co-assay protocols can be hard to implement (e.g., RNA may degrade during sample collection), and it is also impossible to add more experimental modality to existing scHi-C data. Therefore, it is valuable to explore the best strategies to embed the scHi-C contacts. Our focus is to benchmark how each tool represents the single cell contacts so that cell populations are distinguishable in the embedding space. We use co-assay datasets because their co-assayed modalities provide a reliable ground truth of cell identities.

One obvious approach to use the co-assay modality is to embed the co-assayed scRNA-seq or scMethyl-seq data alone and directly place the scHi-C cells in the embedding space of scRNA-seq or scMethyl-seq. Alternatively, the architecture of *Higashi*, *Fast-Higashi*, and *SnapATAC2* allows co-embedding. This approach essentially concatenates the multi-modal data from the same cell and embeds the scHi-C contacts together with their paired modality. It is important to understand that all noises and biases from the scHi-C component are also appended to every cell. Therefore, we still need to decide on the best options to represent the scHi-C contacts. If scHi-C data is not properly represented, the inherited noises and biases may hurt the co-embedding and cause errors. In the co-embedding analyses, the scHi-C and the paired data type will affect each other's embedding: cell clustering from the co-embedding is usually better than clustering scHi-C modality alone, but can be either better or worse than clustering with paired RNA or DNA methylation only, depending on the contribution from the scHi-C modality. A critical point is that the question about scHi-C data representation only pertains to the scHi-C modality. Therefore, we can still focus on embedding the scHi-C component only, regardless of the co-assayed modality. The conclusions from our current benchmark are applicable to all downstream tasks, including the co-embedding analyses.

Co-embedded multi-modal data may facilitate following up discoveries, such as correlated features and new cell subpopulations. A few previous papers showed the possibility to find new cell states or subpopulations from co-embedding analysis. However, to verify the new populations, the authors had to make specific hypotheses about what these new clusters are, then looked for evidence (such as marker gene expression) to support those hypotheses. In these approaches, the new cluster can be indeed verified when the hypothesis is supported. However, when the hypothesis is not supported, the authors will not be able to tell if the new cluster is false positive or if they did not make the correct hypothesis. It is also not feasible to make hypotheses for every additional cluster in a benchmark study. Taken together, without ground truth, we cannot benchmark the clustering from co-embedding results because there is no appropriate "ground truth" to control the false discoveries. We can no longer use the cell type labeling from the paired scMethyl-seq or scRNA-seq data as ground truth because it will be circular reasoning, as the paired modality is included in the embedding.

### Reporting summary

Further information on research design is available in the Nature Portfolio Reporting Summary linked to this article.

## Data availability

The preprocessed data generated in this study have been deposited in the GEO database under accession code GSE305523. Raw data for each dataset are publicly available on the 4DN data portal or at the following Gene Expression Omnibus (GEO) accessions: Mouse

Oocyte/Zygote (GSE80006), Mouse Embryo (GSE129029), mESC Cell Cycle (GSE94489), mESC Cell Cycle scCAREseq (GSE211395), Ramani sciHi-C Mixture (GSE84920), Kim sciHi-C Mixture (4DNESUE2NSGS, 4DNESIKGI39T, 4DNES1BK1RMQ, 4DNESTVIP977, 4DNES4D5MWEZ) [https://data.4dnucleome.org/publications/b8075496-ce51-42a8-8e6c-8cd15a91e353/#overview], Human PFC snm3cseq (GSE130711), Mouse Brain HiRes (GSE223917), Mouse Hippocampus snm3Cseq (GSE213262), Human Brain Atlas (GSE215353). Source data are provided with this paper as Source Data files. Source data are provided with this paper.

## Code availability

All analysis was done with our SCORE Python package which can be installed from https://github.com/JinLabBioinfo/SCORE (a specific version for this publication is archived on Zenodo: https://doi.org/10.5281/zenodo.16878320)[71]. Code for *Va3DE* can be found from https://github.com/JinLabBioinfo/Va3DE (specific version for this publication is archived on Zenodo: https://doi.org/10.5281/zenodo.16878325)[72]. *Va3DE* is also included in the SCORE Python package.

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

## Acknowledgements

This work was supported by grants from NIH R01HG009658 (F.J., J.L., and Y.L.), R01HG012384, UG3NS132061 (F.J and Y.L.), R01CA267872 (F.J.), R01DK113185, R01DK131437 (Y.L. and F.J.). F.J. is also supported by a subaward from NIH project (U01AG072579) through University of Miami and partially by NIH grants R01CA252224, R01GM148662, and P01CA272161. J.L. was also supported in part by NSF CCF-2200255, NSF CCF-2006780, NSF IIS-2027667 and NIH U01AG073323. Y.L. is also partially supported by NIH grants R01CA264320. D.P. is supported by a training grant in Computational Genomic Epidemiology of Cancer (CoGEC, T32CA094186). This work made use of the High-Performance Computing Resource in the Core Facility for Advanced Research Computing at Case Western Reserve University.

## Author contributions

F.J., J.L., and Y.L. conceived the project and secured funding for this project. D.P. developed the *Va3DE* tool and performed the benchmark analysis. X.L. and S.Z. helped with raw sequencing processing and part of the data analysis. D.P. and F.J. wrote the manuscript. X.L. and S.Z. also contribute to the manuscript writing. F.J., J.L., and Y.L. also provided critical feedback and interpretation of results, as well as helping to revise the manuscript.

## Competing interests

The authors declare no competing interests

## Additional information

**Supplementary information** The online version contains Supplementary material available at https://doi.org/10.1038/s41467-025-64186-4.

