## [Transparent Peer Review file · Nature Communications]

A comprehensive benchmark of single-cell Hi-C embedding tools

Corresponding Author: Dr Fulai Jin

Version 0:

Reviewer comments:

Reviewer #5

(Remarks to the Author)

I have read carefully the revised the manuscript, the two rounds of reviews from Reviewer 1, and the response from the authors, it is clear that the authors did not fully address Reviewer #1's concerns. In addition, I also feel the comments from Reviewer 1 are quite reasonable, and I agree with Reviewer 1 on these remarks.

(Remarks on code availability)

Version 1:

Reviewer comments:

Reviewer #2

(Remarks to the Author)

Reviewer #1: "For the embedding evaluation itself, as pointed out in the previous review, the authors did not look closely at the results by simply relying on cell type annotations from other modalities as "ground truth". This is hardly what actual analysis would desire, and the authors did not address this."

There appears to be some misunderstanding going on here. I honestly am not sure I understand what the reviewer is suggesting ("Embedding methods for scHi-C are useful, but they need to be placed in the context of datasets and more meaningful analysis goals."). The authors seem to interpret this as the reviewer asking them to concatenate the two modalities in the embedding space ("In the co-embedding, the scHi-C component will be appended to the co-assayed scRNAseq or scMethyl-C data for each cell."), but this makes no sense to me.

In general, I think Fig. 1a already includes all possible datasets with reliable cell type identifiers. In particular, I think the co-assay datasets are a very reliable source. If the reviewer thinks the ground truth is problematic, then I think the onus is on them to make specific suggestions and some alternative ground truth.

Reviewer #1: "However, the manuscript tends to make broad generalizations without controlling and recognizing the limitation of the dataset being used and the confounding from the variation of scHi-C data/technology itself."

I agree this is a valid concern, but I don't see any reliable way to eliminate such technical variation. In the absence of such a method, using as many datasets as possible is a good way to eliminate the bias.

Reviewer #1: "The premise of this evaluation is questionable given its narrow scope and limited impact. ..."

This point is reasonable, though again the reviewer would be on stronger ground if they made more concrete suggestions. I

would suggest that comparing pseudo-bulk vs. bulk Hi-C might be a useful additional way to evaluate clustered cell types.

Reviewer #1: "2. The authors used several different types of datasets, some older and some more recent...."

I agree that the conclusions are somewhat "over-claimed" here. But considering the limited availability of scHiC datasets, this may be the best the analysis one can do. It is almost impossible to remove such technical biases. In their response, the authors stood by their original claims, but I think they would do better to simply weaken their conclusions here somewhat, admitting more directly the limitations of any benchmarking study in this field.

Reviewer #1: "3. The evaluation remains simplified to some clustering-based metrics, in particular ARI score. ..."

This is a valid concern about over-reliance on the ARI measure. The study would be more convincing if additional performance measures were included in Fig 2-6. Normalized mutual information, Fawkes-Mallows index, purity, homogeneity, completeness, and the V-measure are all ways to get a more nuanced view of the performance. I'm not saying you need to compute all of these, but careful thought to what measures give a complete picture of the performance of the clustering would be beneficial.

Reviewer #1: "4. In the previous round of review, I raised the concern about whether the cell types defined by RNA or methylation can be treated as "ground truth". ..."

This concern is valid; however, personally I think the other modality's clustering is typically more reliable than scHiC since their resolution is much higher than scHiC. In general, it would of course be nice if the authors could explore the scHiC embeddings more carefully, but I don't know what such an analysis would look like and Reviewer #1 didn't make any specific suggestions.

Reviewer #1: "5. There are a lot of details that are missing, ..."

This was a valid concern, which I think the authors have now addressed.

Reviewer #1: "6. The hyperparameter settings used by the authors for their own method and other baseline methods were not explained. ..."

Again, these are valid concerns which have been well addressed in the revision.

Reviewer #1: "7. The claim that methods such as scHiCluster and Fast-Higashi need to construct the entire $N \times M \times M$ matrices and hence lead to out-of-memory error is incorrect. ...

8. On our end, both scHiCluster and Fast-Higashi can run on the complete human PFC dataset (4k cells) at 100Kb resolution consuming less than 40GB of memory with recommended parameters. "

The authors appear to have addressed this concern.

Reviewer #1: "9. The last section on integration based on scGAD is very weak. ..."

At this point, the reviewer re-raises the point about co-assays, and here they more clearly state that they want the authors "to expand the evaluation on co-embedding with such paired co-assay data with scHi-C." I think the authors have already used the co-assay data in the right way, namely, as a source of orthogonal gold standard labels of cell types. Further asking them to actually do co-embedding of the two data types doesn't really make sense to me, since as the authors correctly point out, you then don't have any gold standard. More generally, I anticipate that developing a co-embedding method for scHi-C and other, linear assay would be quite challenging and is clearly beyond the scope of this paper.

Reviewer #1: "10. I read the authors' responses carefully. Here are my responses. ..."

I think the authors have responded well to all of these points, with the exception of point d. It's certainly possible to do downsampling to benchmark imputation. However, I don't think adding this experiment would add much value to the paper.

(Remarks on code availability)

We are sorry that reviewer #1 is not satisfied with our previous responses. We addressed most of the issues brought up previously, but the reviewer insists on the remaining criticisms related to the scopes of our study. Specifically, although our study is largely a benchmark of scHi-C data embedding, the reviewer wanted us to add new benchmarks on "downstream tasks" such as "new cell type discovery" and "single cell imputation".

In our previous responses, we mentioned that we could not benchmark those tasks due to the lack of "ground truth" to do so, but the reviewer disagrees with our responses. He/she further elaborates those criticisms and questioned the significance of our study. We have now provided a detailed explanation why it is not feasible to benchmark the "downstream tasks" as the reviewer requested.

It is also important to point out that "embedding" is about how scHi-C data should be represented in the computer. It is therefore fundamentally important and the conclusions from our benchmark study are applicable to all "downstream tasks".

The reviewer made a few more comments related to our embedding benchmarks. We have addressed those additional issues below.

Below the reviewer's comments are in **RED**, and our responses are in **BLACK**. For convenience, we have shaded our responses related to the reviewer's request on "downstream tasks" in **GREY**.

Reviewer Comments:

Reviewer #1 (Remarks to the Author):

I appreciate the additional work from the authors and their responses. However, most of my original concerns remain, and the revisions fail to address significant questions. Upon closely examining the provided code and evaluations, I find major issues with the credibility and rigor of this benchmarking study. Although scHi-C data embedding is a key step for single-cell Hi-C analysis, the authors' oversimplification of the analysis and reliance on problematic metrics undermine the value of this work. The overemphasis on embedding and the unwarranted generalizations drawn from it demonstrate some fundamental misunderstanding from the authors on the broader context of single-cell 3D genome research.

For the embedding evaluation itself, as pointed out in the previous review, the authors did not look closely at the results by simply relying on cell type annotations from other modalities as "ground truth". This is hardly what actual analysis would desire, and the authors did not address this. Moreover, the authors claim that this evaluation intends to assess the performance of different scHi-C embedding methods. However, the manuscript tends to make broad generalizations without controlling and recognizing the limitation of the dataset being used and the confounding from the variation of scHi-C data/technology itself. Given the lack of consideration of the technical variation itself, the limited scope of the evaluation, and the lack of transparency and rigor of the evaluation itself, this work falls significantly short on multiple fronts. My detailed comments are elaborated below.

We are sorry that reviewer #1 was not satisfied with our previous responses.

Firstly, the scope of our study is to benchmark "embedding", which is about how the scHi-C data should be represented in the computer. For scHi-C, embedding involves the choices of resolution, loop size, preprocessing options such as feature selection, random-walk, IDF, and dimension reduction options with or without deep-learning *etc.* Existing tools use very different options representing the scHi-C data, and our focus is to evaluate the impact of these options. Our work is novel because a comprehensive comparison of these options is still missing from existing literature. Our work is significant because our findings apply to all downstream tasks including but not limited to clustering.

In fact, most of the previous studies only pick a few default options with little consideration that how data representations may change the outcomes. We demonstrated that for many tools just changing the data representation can significantly improve their embedding performance. In this study we primarily use clustering analysis to assess the embedding quality as it is the most reliable and effective approach.

However, our conclusion applies to all other “downstream tasks”, including but not limited to clustering. We now added a clarification to the first paragraph of the discussion section.

Additionally, we also want to point out that the novelty of our study includes *Va3DE*, a new tool for scHi-C embedding. Interestingly, although *Va3DE* only uses a “conventional” CNN model, it is a top performer among all tools, especially for embeddings at high resolution with large cell number.

The reviewer challenged us to explore more “downstream tasks” such as finding new cell states or populations from scHi-C data. We will discuss more detail below that it is not feasible to benchmark those tasks without a ground truth. In literature, those “downstream tasks” were performed as exploration efforts, although it is possible to verify individual cases based on prior knowledge, this approach does not support a rigorous benchmark study due to the lack of ability to determine false discoveries.

We respectfully disagree with the reviewer’s comments about technical variations may affect our conclusions. We will explain below how we have carefully controlled each embedding options. Furthermore, our benchmark datasets coming from very similar “C-” chemistry, no known biases exist between the techniques.

Finally, the reviewer has some new technical comments, especially regarding hyperparameters and memory usage. We have provided additional information and clarified these issues.

1. The premise of this evaluation is questionable given its narrow scope and limited impact. Evaluating scHi-C embedding methods in isolation has little practical utility without considering the biological and technological contexts.

[scope issue] As summarized above, we are benchmarking “embedding” which has fundamental importance beyond cell type annotation or clustering. Embedding is about how the scHi-C data should be represented in the computer, which involves the choices of resolution, loop size, feature selection, preprocessing (such as random-walk and IDF), dimension reduction (with or without deep-learning), etc. Every tool must choose how to represent the scHi-C data, but current literature lacks a systematically evaluation how these options impact the embedding quality. In fact, we demonstrated that the performance of a tool can be significantly improved by changing the data representation. Benchmarking the embedding is therefore fundamentally important and our conclusions apply to all “downstream tasks”.

The variability in “performance” across datasets, as presented in this study, may reflect more on the diversity of technologies and experimental conditions than on the embedding methods overall. Metrics like ARI, while commonly used, are insufficient to capture the nuances of scHi-C data analysis.

All scHi-C are based on the same “C-” chemistry and have been shown to be able to capture chromatin interactions at all distance ranges. We respectfully disagree with the reviewer’s comments about technology diversities and please see more discussion in our response to comment #2.

Please also see more discussion about using ARI in our response to comment #3 below. In fact, we only need to make slight modifications to ARI to robustly measure embedding quality in a wide-range of applications. Specifically, we have introduced a modified “neuron ARI” to measure the embeddings of neuron subpopulations.

The authors’ claim that scHi-C data’s primary purpose is to cluster cell types is misleading and reductive.

This is a misunderstanding. Our exact languages are:

In our previous point-by-point: “...if the goal of a single cell study is to analyze cell populations in complex tissues or cell mixes (which is perhaps the most common application), good embedding and clustering are more important...”

In the methods: “...the goal of most single cell studies is to analyze cell populations in complex tissues or cell mixes, ...”

In these sentences we are talking about analyzing cell populations in complex tissues, not just clustering cell types as the reviewer understood.

As has been demonstrated previously, cell types are more reliably defined by scRNA or scMethylations with a caveat that they are not necessarily ground truth as new cell identity can be complemented by scHi-C. Embedding scHi-C data can assess the quality of the data and reveal cell states not represented by RNA, methylation, or other molecular features. In addition, as discussed in the field of single-cell epigenomics, scHi-C has the potential to reveal individual cell variation of various types of 3D genome features and also understand their roles in gene regulation and other cell functions. Sure, scHi-C embeddings can separate the individual cells before being grouped into pseudobulk, but that's just one type of analysis, which in itself does not really care about the minor differences in clustering as long as the groups are reasonably separated.

We use the cell type labeling from co-assayed scRNA-seq or scMethyl-C data to benchmark the scHi-C clustering. The reviewer argued that the labeling may not represent true cell identity (or true positives).

Firstly, for a benchmark analysis, the ground truth does not necessarily have to be true positives. In fact, true positives are often unavailable in benchmark studies. It is a common practice to use orthogonal data as ground truth looking for better consistency.

Second, our benchmark goal is to seek the optimal settings to represent scHi-C data. The cell type labeling we used is adequate to serve this goal because it has allowed us to compare various data representation options and determine their pros and cons under different circumstance. These knowledges have fundamental important for all scHi-C analyses. For example, in the task of “reveal cell states not represented by RNA, methylation”, it is still important to choose the best options to represent scHi-C data.

[scope issue] Third, the reviewer brings up the task to reveal new cell states based on scHi-C data. Unfortunately, although this task is interesting, it cannot be rigorously benchmarked due to the lack of ground truth. Using scHi-C to reveal “new cell identity” beyond what can be shown by scRNA or scMethylation is very risky because scHi-C data is prone to noises and biases. The priority in such effort is to exclude the possibility of false discovery. Although past literature demonstrated the possibility to achieve this, their verification approach requires hypothesis generation. Such hypothesis can verify a new cell state if tested true but does not determine false discovery if tested false. Therefore, it is not feasible to perform a rigorous benchmark as this reviewer requested. We have more detailed discussion about this point in our responses to comments #4, #9, #10 below.

Importantly, the growing list of experimental technologies and datasets with co-assays in the past two years points to a direction of analyzing multiple modalities from the same cell, including scHi-C. Embedding methods for scHi-C are useful, but they need to be placed in the context of datasets and more meaningful analysis goals. I raised this suggestion in the previous round of review and encouraged the authors to evaluate other components of scHi-C analysis; at least assess the impact of embedding in downstream tasks. The authors, unfortunately, did not devote effort to address this. This study's overemphasis on minor differences in clustering metrics, without connecting them to practical outcomes, significantly diminishes its relevance.

[scope issue] In the co-embedding, the scHi-C component will be appended to the co-assayed scRNA-seq or scMethyl-C data for each cell. Cells are embedded with the paired data. It is important to understand that all noises and biases from the scHi-C component are also appended to every cell. Therefore, the fundamental importance of scHi-C embedding should not be down-played. We still need to decide on the best options to represent the scHi-C contacts. If scHi-C data is not properly represented, it will hurt the co-embedding and cause errors.

A critical point is that the question about scHi-C data representation only pertains the scHi-C modality. Therefore, the conclusions from our current scHi-C embedding are applicable to all downstream tasks, including the co-embedding analyses.

Unfortunately benchmark co-embedding results is not feasible due to the lack of ground truth. A few previous papers showed the possibility to find new cell states or subpopulations from co-embedding analysis. However, to verify the new populations, the authors had to make specific hypotheses about what these new clusters are, then looked for evidence to support those hypotheses. In these approaches, the new cluster can be verified when the hypothesis is tested right. However, when the hypothesis is tested wrong, we cannot tell if the new cluster is false positive, or if we just need another hypothesis. This is a major issue for benchmark study because we cannot determine if a new cluster is false discoveries. Taken together, even if a “downstream task” can be verified in an explorative study, it is still not feasible to benchmark without ground truth. We have more discussion about this issue in our responses to comments #4, #9, #10.

We added more clarification about co-embedding from above into the “additional discussion” section.

2. The authors used several different types of datasets, some older and some more recent. It is important to point out that these datasets were not generated by the same scHi-C technology. Even with the same scHi-C assay, there are major variations among different datasets given how rapidly these methods evolve. These datasets also concern different biological contexts. The authors, unfortunately, made very broad conclusions (or perhaps just mostly speculations) about the embedding evaluation results. For example, certain cell contexts may prefer long-range or short-range contacts from scHi-C. However, the authors did not control the potential confounding technical variation of these datasets. Different technologies have different capabilities of capturing long-range contacts for instance. Therefore, the variation of contributions in different ranges of contacts could contribute to the technical variation, dataset batch variation, rather than actual biological variation. The authors did not carefully evaluate this, which is fundamentally important, especially when making claims from the embedding results. Without controlling for these confounders, the authors’ broad conclusions about embedding performance are speculative at best. This lack of rigor undermines the validity of the claims.

We respectfully disagree.

Firstly, we intentionally included different types of datasets to seek general conclusions that are independent of technologies; this choice should not be considered as a strength, not weakness. These benchmark datasets are frequently used in literature for benchmark analyses and for tool development. Furthermore, there are no alternative scHi-C datasets for some biological questions, such as early embryo and cell cycle.

Secondly, all the single cell Hi-C technologies are based on the same C-chemistry, their capabilities to capture both short- and long-range contacts are well established in the literature. There is no evidence indicating that these technologies have different capabilities capturing long-range contacts.

Thirdly, we have carefully controlled the embedding options in our comparisons. In many cases, we also include additional evidence to support our conclusions, too. To be more specific:

- We concluded that certain cell contexts may prefer long- or short-range contacts based on a series of analyses in **Fig.2-4**. In these tests, we always compared the results when only long- or short-range contacts from the same dataset are used. Therefore, the variation between datasets do not affect these conclusions.
- We also performed additional pseudo-bulk data to support our conclusion that “certain cell contexts may prefer long-range or short-range contacts”. For example, **Fig. 5h** shows significant compartmental differences between neurons and non-neurons at long-range, but the compartments cannot distinguish neuron subtypes.

- Note that the mouse early embryogenesis data and cell cycle data are generated with the same scHi-C protocol. We found that mouse early embryogenesis data prefer long-range contacts and cell cycle data prefer short-range contacts. This example also shows that our conclusion about distance preference is not due to protocol variation.

3. The evaluation remains simplified to some clustering-based metrics, in particular ARI score. However, the UMAP typically shows minor differences even when the ARI scores are different. This relates to the concern that the authors mostly try to quantify the embedding results using the details of the cluster metrics but do not analyze the actual implications in downstream analysis. For example, the UMAPs in Figure 2, Figure 3, and Figure 4, the authors argue that the different loop contact sizes have an impact on embedding results. However, even though the ARI scores show some differences, the actual UMAPs typically do not demonstrate major differences where the clusters can be clearly separated or the biological processes can be separated. In addition, in Figure 5, the authors made an attempt to assess the impact on random walk, but the UMAP results show less obvious differences even though the ARIs are reported to be different. This extends to the method evaluation for downsampling data in Figure 6 where most of the downsampling results look like the clusters can be separated for different methods even though the ARI scores are dropping. Overall, this raises the question regarding the significance of the evaluation, in particular, using metrics such as ARI and related scores designed for clusters for this evaluation.

Many of the panels in **Fig. 2-6** are highlighting the separation of cell subpopulations in the same cell clusters, many are relevant to the neuron subpopulations in the brain scHi-C datasets.

For example, in **Fig. 3a-c** (shown below), we found that with InnerProduct (first row UMAPs), the 4 excitatory neuron subtypes and the 4 inhibitory neuron subtypes are recognizably separated inside the two big neuron clusters when using <2Mb or using <20Mb contacts (**Fig. 3a**, panels in top row), but the subtype differences are not observable when using >2Mb contacts (**Fig. 3b**, top panel). For scHiCluster, these subtypes are recognizable when only <2M contacts are used, inclusion of long-range contacts >2Mb hurts the embedding (**Fig. 3a-b**, panels in bottom row).

Figure 3a-c copied from the main figures.

We have a few points to make here in response to reviewer's comments about ARI.

- ARI can robustly measure the variances both between clusters and within clusters. In the example of **Fig.3a** above, if we enforce a high number of clusters, subclusters may form within the two big neuron clusters. If embedding is good, the subclusters will better reflect the neuron subtypes and yield higher ARI.

- In the example of **Fig. 3c** shown above, we are using “neuron ARI” to only focus on neuron subtype separation. The “neuron ARI” measurement agrees very well with how neuron subtypes separate in t-SNEs of **Fig. 3a-b**. We also used “neuron ARI” in **Fig. 3f,5g,6c,6g**.
- Many of the panels in **Fig. 2-6** demonstrated how changes of embedding options affect the in-cluster differences including: **Fig. 4e-f**, cell cycle subpopulations among the 64C embryo cells; **Fig. 6b-c**, 8 neuron subtypes in the human PFC dataset; **Fig. 6e-f**, 19 neuron subtypes defined in the human brain atlas data. In all these examples, the ARI or neuron ARI can robustly measure the separation of cell subtypes.

Regarding the significance to benchmark scHi-C embedding, *i.e.*, why is it important to evaluate the quality of scHi-C embedding if the cell subtypes still cannot be clearly separated into discrete cell clusters?

- It is important to understand that our goal is to benchmark embedding, *i.e.*, we seek the best options to represent scHi-C data. This is a meaningful goal because our findings are applicable to all “downstream tasks”, including but not limited clustering.
- Although neuron subtypes in our benchmark datasets cannot be clearly separated with existing tools, the knowledge we obtained from these analyses (about the optimal embedding options) will help the clustering of new datasets or develop better embedding tools.
- Optimal embedding options are important for all other “downstream tasks”, including some tasks mentioned by reviewer #1: (i) Co-embedding scHi-C/scRNA-seq or scHi-C/scMethyl-C when paired modalities are available; (ii) When paired modalities are unavailable, joint embedding the scHi-C data with independent scRNA-seq data, as we discussed in **Fig. 7**, can be extremely helpful; (iii) Impute genome architecture in single cells; (iv) Trajectory/pseudo-time analysis of genome architecture. In all these cases, we need to make choices to represent scHi-C data, such as resolution, distance range, *etc.*
- Finally, evaluating scHi-C embedding is in many ways analogous to evaluating different AI models, such as “ancient” attention models and “modern” large language models such as ChatGPT and DeepSeek. Different AI models can be considered as different embedding of world knowledges (text, image, videos, *etc.*). Improving the embedding has fundamental impacts on all downstream applications.

4. In the previous round of review, I raised the concern about whether the cell types defined by RNA or methylation can be treated as “ground truth”. The authors ignored that question and argued that one has to have a ground truth. I actually disagree with this. In a way, these cell types “ground truths” are also inferences from other modalities. Whether they capture all the cellular complexity in a certain context is unclear. As in previous work such as the snm3C-seq work to demonstrate the new cell cluster and states defined by scHi-C, the authors should have taken the cell types defined by other modalities with a grain of salt and explored the clusters from scHi-C embedding more carefully. The current evaluations are somewhat biased, and it is unfortunately unclear how well each method reveals biologically interesting cell states.

We claimed that a benchmark study must have a “ground truth”. Here the reviewer argued that the “ground truth” we used may not be “true positive”. However, the reviewer’s arguments do not change the fact that we still need a “ground truth” for a robust benchmark study. As discussed above already, in a benchmark analysis, “ground truth” does not necessarily have to be a “true positive”. In fact, “true positive” is often unavailable in benchmark studies. The common practice is to use orthogonal reference data as ground truth reference, which is what we did in this study.

[scope issue] Here, the reviewer again mentioned the “downstream task” to use scHi-C to find new cell states. For example, in the last round of review, the reviewer mentioned the Fast-Higashi paper (Zhang et al., PMID:36265466). In (Zhang et al., Fig. 4a-b), the authors observed two small clusters of interneurons from scHi-C data that are unannotated in the original study. To validate their observation, Zhang et al. compared the activity of Pvalb, Sst, and Vip genes and determined that one cell cluster is Pvalb/Sst

positive interneurons, and the other is Vip positive interneurons. Please note that here the authors had to make hypothesis that Pvalb, Sst, and Vip can distinguish these cell clusters based on prior knowledge.

Although this example shows the potential of scHi-C in “discovering new cell states” in specific cases, we cannot generalize this phenomenon that all new clusters from scHi-C data are real biology, especially because scHi-C data are prone to noises and biases. In the example discussed above, the authors had to make specific hypothesis to verify the new cell clusters based on prior knowledge. In this approach, if the hypothesis is not supported, because we do not have a ground truth, it will be impossible to tell if the new cell cluster is a false discovery, or if we just need another hypothesis. This dilemma makes a benchmark study impossible because we cannot control the false discoveries. Furthermore, it is not feasible to make hypothesis for every additional cluster. We have the same discussion related to this issue in response to comment #1, #9, and #10.

5. There are a lot of details that are missing, raising concerns about the credibility of the study. It is unclear about the number of cells in the human brain atlas dataset. The original Tian et al. dataset contains more than 140k cells, but only 32k cells were used. Both scHiCluster and Fast-Higashi have tutorials for running their respective algorithms on the complete dataset and achieve good results. It is unclear why the authors only used part of the dataset.

<https://zhoujt1994.github.io/scHiCluster/hba/intro.html>

https://github.com/macompbio/Higashi/blob/main/tutorials/FastHigashi_m3c_Tian_HumanBrain_500k_part1.ipynb

https://github.com/macompbio/Higashi/blob/main/tutorials/FastHigashi_m3c_Tian_HumanBrain_500k_part1.ipynb

This is because when we were preparing the manuscript for the first submission, only a subset of this dataset were available for download. We later decided to keep using 32K cells as this number is still very big among current scHi-C literature, and it is big enough to support our conclusions. If we use the entire 140K-cell datasets, more tools will fail, making it harder for us to compare.

6. The hyperparameter settings used by the authors for their own method and other baseline methods were not explained. We looked into the code shared on GitHub and it's difficult to assess whether the evaluations were conducted fairly and how they were carried out.

6a. The authors briefly mention they used “recommended” or “default” values, but since methods and their documentation are updated over time, it is crucial to explicitly report the parameters used to achieve a fair comparison.

We added a page of hyperparameter usage on the GitHub.

<https://github.com/JinLabBioinfo/SCORE/tree/main/tutorials>

6b. Related to the above point, for Fast-Higashi, the authors did not use the recommended “filter=True” option. Meanwhile, Fast-Higashi also has a guideline to select hyperparameters according to resolution and coverages, but it seems the author kept it as the same for all experiments, which seems undesirable for the practical utility of the method and unfair for the analysis.

In *Fast-Higashi*, filter=true does additional cell filtering based on its own QC criteria, *i.e.*, more cells will be filtered out before embedding. For a fair benchmark we need to be clustering the exact same cells across embedding methods. We therefore choose to use a consistent filtering criterion across all method sweeps.

We avoid changing the parameters of a tool when switching between different datasets so that the same embedding option can be evaluated across different datasets.

Since our goal is to compare the embedding options of scHi-C data, we only change a parameter when we want to explore how the change may impact the results.

6c. For the authors' method Va3DE, did they use different hyperparameters for each dataset (e.g., number of Gaussian mixtures)? If so, this raises the question of whether the comparisons are fair.

We used 30 Gaussian mixtures for all complex tissue datasets and use 10 mixtures for the other smaller datasets, both are overestimation of actual number of clusters. We have reported this information in a hyperparameter page. Based on our analysis in **Supp Fig 13** we concluded that using more Gaussian mixtures does not affect the results.

6d. Very importantly, what are the latent dimensions or final cell embedding dimensions used, and how were they determined? This information is missing in the paper. Based on the GitHub repository, it seems a dimension of 32 was used for all methods and datasets. However, the optimal dimension can vary significantly across methods and datasets (e.g., depending on cell type composition complexity). Deep learning methods with more layers can potentially preserve more information with lower dimensions, while simpler models might require higher dimensions. Additionally, whether changing cell embedding dimensions requires retraining is an important factor. For example: In scHiCluster and Fast-Higashi, embeddings are ranked like principal components (PCs), with the most important ones coming first. This makes it straightforward to run models with higher dimensions and later decide on a lower dimension. In contrast, deep learning-based methods such as Higashi and scVI-3D require retraining when dimensions are adjusted. It's unclear how the authors ran different methods, raising further concerns about whether the evaluation was conducted thoughtfully.

We expect that an ideal embedding tool should have a robust dimension reduction algorithm that is applicable to multiple datasets. End users would also prefer choosing a constant dimension rather than tweaking the number of dimensions. In this study we keep a constant number of 32 dimensions. This number is adequate because with 32 dimensions there is at least some method which achieves satisfactory performance on each dataset.

6e. When doing clustering using Louvain, does the author use Euclidean distance or cosine distance? It's less intuitive why the authors chose cosine only for the silhouette coefficients.

We use the scanpy implementations of Louvain and Leiden which use Euclidean distance by default. However, for silhouettes we use cosine because different methods produce embeddings with different geometries and scales of Euclidean distances. The silhouettes are more affected by outliers when using an unbounded distance like Euclidean but more consistent and comparable between methods when using cosine distance.

6f. For Figure 1, it is unclear if all strata were used or only a subset of the strata is used for all methods.

Each tool has its own default numbers of strata. ScHiCluster uses all strata. Most tools use 32 strata. Higashi uses 60 and fast-Higashi uses 100 strata. ScGAD does not have this option as it only analyzes contacts on genes. This information is now listed in the hyperparameter page in the SCORE GitHub.

7. The claim that methods such as scHiCluster and Fast-Higashi need to construct the entire $N \times M \times M$ matrices and hence lead to out-of-memory error is incorrect. These matrices are stored in sparse format in both software (and supports `n_strata`), and tensor decomposition / SVD operates directly on the sparse matrices. While some implementations may achieve better or worse memory optimization which could lead to out-of-memory error, it is misleading to provide incorrect analytical memory consumption analysis. It is important to separate conceptual advancements and software development improvements.

8, On our end, both scHiCluster and Fast-Higashi can run on the complete human PFC dataset (4k cells) at 100Kb resolution consuming less than 40GB of memory with recommended parameters. It is unclear to me why both methods are reported as out-of-memory. Again, without reporting the detailed hyperparameters, it is hard to assess the reason. The authors need to be much more transparent on this. As mentioned above, it is certainly not due to constructing the full NxMxM matrices as both methods operate on the sparse data.

Comments #7 and #8 are about the same issue and are addressed together.

Although fast-Higashi indeed stores data matrices in sparse format, during initialization it does random-walk and attempts to generate a full NxMxM matrix for each chromosome. See

https://github.com/ma-compbio/Fast-Higashi/blob/71182a9bdee2b96cd1676e1448285c5705a354e7/fasthigashi/sparse_for_schic.py#L291

https://github.com/ma-compbio/Fast-Higashi/blob/71182a9bdee2b96cd1676e1448285c5705a354e7/fasthigashi/parafac2_intergrative.py#L124

We met out-of-memory issues at this random-walk step. Random-walk is enabled by default although one can disable the random-walk option to bypass this step during initiation, but this is only recommended for “high quality scHi-C data”. We suspect that the reviewer disabled random-walk in his/her own tests. However, since partial random walk is a key innovation of the Fast-Higashi pipeline, (it is highlighted in the abstract of the Zhang et al. paper), we kept it on across all benchmark datasets. We have added explanation in the methods section. We also have added a sentence in the main text about the random-walk option.

“...we found that memory performance of *fast-Higashi* can be improved by disabling its random-walk step, noting that the authors recommend this option for high quality scHi-C data with more contacts per cell...”

As for scHiCluster, although it also stores data matrices in sparse format, when doing PCA for each chromosome it must convert the data into dense matrices, which increase the memory usage to the NxMxM scale. The latest version of scHiCluster has the `n_strata` option that can decrease memory usage by only using contacts within a certain size range (e.g., 0-2Mb). This option decreases the number of values we must load into a dense matrix before computing principal components, which is likely how the reviewer gets around the memory issue. But using `n_strata` is different from the published scHiCluster algorithm that we are benchmarking. The original algorithm is well-suited for long-range contact embedding, as we showed in **Fig. 2-3** that for scHiCluster, early embryonic data relies on long-range contacts.

Importantly, the goal of our study is to benchmark the embedding options seeking the best option to represent scHi-C data. Because both random-walk and `n_strata` are considered as embedding options, in **Fig. 2-5** we have systematically analyzed how these options affect embedding performance across different tools, although we did not change the random-walk and `n_strata` options in the initial benchmark in **Fig. 1**.

We have added some clarifications about scHiCluster and fast-Higashi in the “Method Memory Complexity Details”.

9. The last section on integration based on scGAD is very weak. First, the issue is that scGAD ignores the vast majority of scHi-C contact information while only focusing on gene body contacts. How biased and unreliable this is remains to be explored. In our hands, the method seems to work okay for brain tissues that are easily separated with their cell types but does not work well for other cellular contexts.

I don't think we are saying anything different here. Yes, scGAD is not a strong performing tool. We already pointed out in the paper that it does not work well with early embryogenesis and cell cycle datasets, and it

work averagely with complex tissue data. The total rank of scGAD is low, presumably because it only use gene body contacts for embedding.

In the last section, **Fig.7** aims to show that for scGAD, integration with independent scRNA-seq can dramatically improve the results compared to embedding scHi-C alone. We feel that it is necessary to give scGAD a fair credit because it is originally designed to facilitate integration with scRNA-seq data.

In addition, there is a growing list of technology developments in the field to profile scHi-C together with other modalities, especially RNA. I raised the suggestion to expand the evaluation on co-embedding with such paired co-assay data with scHi-C. However, the authors ignored this suggestion. In their response, the authors argued that co-embedding cannot be evaluated given that there is no ground truth. This is hardly the case. I urge the authors to revisit the notion of “ground truth”. In such a cell typing effort, it’s often the case that there is no ground truth per se but need careful analysis of the identified clusters themselves to evaluate the results. This was explored for earlier co-assays such as snm3C-seq and more recent scHi-C and RNA co-assays. I think this is of great importance to the field to not only assess individual cell variation of these modalities but also understand their dynamics in complex cell contexts. The unpaired integration, as the authors did using scGAD, is limited in its utility for many reasons mentioned above and will not reveal much new insights.

[scope issue] Here the reviewer brings up the cope issue again asking us to add a benchmark of co-embedding results on paired scHi-C/scMethyl-C or scHi-C/scRNA-seq co-assay data. We discussed about the difficulty to benchmark this task in our response to comments #1 and #4. Here we summarize the previous points again and add a few more points.

- Our benchmark of embedding seeks the best options to represent scHi-C data in computer. This scope only pertains the scHi-C modality. Therefore, it is justifiable do benchmark on the scHi-C modality only. When scHi-C data is co-embedded with scRNA-seq or scMethyl-C data, the evaluation involves the embedding of the scRNA-seq or scMethyl-C modality and the contribution from additional modality, which is no longer about scHi-C data.
- Optimal embedding options are important for all other “downstream tasks”, including the co-embedding task. During the co-embedding, we still need to make choices to represent scHi-C data, such as resolution, distance range, *etc.*
- We use the cell type labeling from the paired scRNA-seq or scMethyl-C data as “ground truth” for benchmark, but in the case of co-embedding there will be no “ground truth”.
- The reviewer questioned about our ground truth and argued that “there is no ground truth per se”. It is important to note that “ground truth” and “true positive” are conceptually different. In a benchmark analysis, “ground truth” does not necessarily have to be a “true positive”. Our approach of using cell labeling from co-assayed modality as “ground truth” is a justifiable common practice.
- The reviewer mentioned previous works claiming that it is possible to verify new cell states without ground truth. We want to point out verification is different from benchmarking. In previous studies, to verify the new populations, the authors had to make specific hypothesis about the identifies of the new cell clusters, then look for additional evidence (such as marker gene expression) to support the hypothesis. In these approaches, although the new cluster can be verified when the hypothesis is supported, when the hypothesis is not supported, we cannot tell if the new cluster is false discovery, or if we need a new hypothesis. There is no way to control the false discoveries which is necessary for a benchmark study.
- We disagree with the reviewer’s assertion that unpaired integration is less useful. Developing co-assay protocols also has its own limitations: (i) the protocols are generally hard to do, sometimes even impossible to do or with questionable quality (e.g., due to RNA degradation); (ii) developing co-assay protocols with 3 or more modalities is even harder; (iii) it is also impossible to add more modality to a large amount of existing data. Due to these reasons, integrating unpaired single cell data can be practically very useful. We agree that method like scGAD has its own weakness. But

by showing its potential with scGAD, we convey the message that it will be valuable to make more efforts on unpaired integration methods.

We added more clarification about co-embedding from above into the “additional discussion” section.

10. I read the authors’ responses carefully. Here are my responses.

10a. The authors wrote: “SnapATAC2 ranks the highest among all non-deep-learning tools. We now recommend SnapATAC2 as a top choice due to this result.”

=> This is perplexing because the main outcome from the authors’ evaluation is that different datasets may need different embedding methods.

The recommendation is based on the overall ranking of SnapATAC2, but SnapATAC2 does not win every time. We found SnapATAC2 works well across most benchmark datasets and works well with low depth data, too. However, we also pointed out that SnapATAC2 does not work very well distinguishing neuron subtypes at high resolution.

10b. I do not agree with the authors’ response on imputation, and the authors disregard this important point. I think the authors did not recognize the actual value of scHi-C data. It’s obviously not just through embedding to identify cell types. Rather, it’s more important to use scHi-C data to identify individual cell variation of 3D genome features and build a strong understanding of genome structure and gene expression.

[scope issue about imputation] We did not disregard the point about imputation. Our last revision included a detailed two-page long discussion about imputation in the “additional discussion” section.

The reviewer asked us to compare the data imputation from Va3DE. In the last revision we have explained already that the variational autoencoders in methods like Va3DE are not suitable to generate stable single cell imputations due to the usage of random distributions in the latent space.

Out of the only three tools (scHiCluster, Fast-Higashi, and Higashi) that can do imputation, the authors of Fast-Higashi have already showed its imputation is nearly identical to scHiCluster. The Higashi authors already compared its imputation to scHiCluster imputation. There is essentially nothing more to do on this issue.

We don’t think we are on different pages about recognizing the value of scHi-C data in transcription regulation. But I would not say studying individual cell variation is “more important” than analyzing the 3D genome features at cell population or subpopulation levels. We still have a long way to go on the mechanistic and functional characterization of differential 3D genome features between an increasing number of cell populations or subpopulations from single cell genomic data. There are a lot of practical needs in this area that has great potential for both basic and translational research. On the other hand, it is also unclear if the cell-to-cell variability observed from scHi-C data, really represent meaningful differences with transcriptional consequences rather than stochastic variability. After all, DNA looping during transcription regulation is a dynamic processes and formaldehyde-based “C”-protocol only capture random “freezing” snapshots in those processes. This debate is certainly out of the scope of the paper.

10c. Higashi is not imputation on pseudobulk data. Operating on the cell embedding space does implicitly borrow information from a small number of nearby cells in the embedding space, but it’s not the same as pseudobulk embedding. I think the authors’ generalization is unfair. Higashi still reveals individual cell variation, and pseudobulk-level imputation will fundamentally obscure that. The authors’ statement that “Higashi may suppress cell-to-cell variations within the same cell population” is not quite true.

Our language was:

...Higashi uses a unique hypergraph data representation to allow borrowing information from k (default 4) nearest neighboring cells for data imputation after embedding is done. Therefore, this approach can be considered as “local pseudo-bulk imputation”...

Here we are making an analog of this approach to pseudo-bulk analysis, but we do realize that this sentence may lead to misunderstandings. We apologize for this and have revised these words to:

...Higashi uses a unique hypergraph data representation to allow borrowing information from k (default 4) nearest neighboring cells for data imputation after embedding is done. Therefore, this approach can be considered as “local pseudo-bulk imputation” except that there is no need to explicitly pool the data from neighboring cells...

In our discussion we discussed that Higashi may still obscure cell-to-cell variation (although to a less extent) because imputation is done after embedding. This is in contrast with scHiCluster imputation (random-walk) which happens before embedding. When imputation is done after the embedding, population level features are already learned in the hypergraph. The Higashi authors also showed in their paper that the imputed data agrees better with a “pseudo-bulk” 3D genome image data. It is therefore not unreasonable to expect that Higashi imputation may display the learned population features from its hypergraph, at least more so than scHiCluster.

10d. The authors argue that they cannot benchmark the imputation of cell-cell variation because there is no proper ground truth. This is hardly the case. One can use higher-resolution data and downsample as several prior works have already demonstrated this for evaluating imputation performance.

[scope issue about imputation] Even the deepest single cell Hi-C data only has <1 million contacts. Imputation to that level is not very useful and hardly what one would wish to do with either scHiCluster or Higashi imputation.

10e. The authors argue that they benchmark different embedding tools, not the experimental protocols. But the authors’ results may indeed reflect technical variation rather than the performance of different tools and certainly not entirely biological implications as the authors stated in the paper.

In our response to comment #1 above, we have discussed above that our conclusions are not due to protocol differences.

10f. The authors stated: “This is not the focus of our benchmark not only because it lacks broad significance, but also because we lack a proper ground truth to do so” and “Most of the scHi-C co-assay protocols were developed because it is hard to cluster the single cells only based on scHi-C modality.” => Both are not true. There are more careful ways to evaluate the results. The perceived ground truth is also an inference/prediction. Also, the co-assays were designed to study the role of the 3D genome on other genomic features in single cells.

[scope issue] As discussed in response to comment #1,#4,#9 above, although new cell clusters from co-embedding analysis can be verified with “more careful ways”, these approaches had to make specific hypothesis about the new clusters and then find evidence, such as marker gene expression, to support the hypothesis. We discussed one such example in the response to comment #4.

Verification is different from benchmarking. In those “more careful ways”, the new cluster can be indeed verified when the hypothesis is supported. However, when the hypothesis is not supported, we will not be able to tell if the new cluster is false positive from scHi-C errors, or if the authors need a new hypothesis. There is no way to control the false discoveries which is necessary for a benchmark study. It is also not feasible to make hypothesis for every additional cluster in a benchmark study

The sentence "...Most of the scHi-C co-assay protocols were developed because it is hard to cluster the single cells only based on scHi-C modality..." was true for snm3C-seq as it is what the original snm3C-seq paper claimed. We acknowledge that as the scHi-C/scRNA-seq co-assays gets more mature, more research topics can be explored relevant to transcription regulation. We have now deleted this sentence.

*10g. The authors stated: "Our focus is to benchmark how each tool represents the single cell contacts so that cell populations are distinguishable in the embedding space. We use co-assay datasets because their co-assayed modalities provide a reliable ground truth of the cell identities."
=> This is not really the case as discussed earlier and in other earlier scHi-C works showing the value of scHi-C defining cellular states/clusters by assessing the unique 3D genome features of that particular additional new cluster, not defined by, say, RNA.*

[scope issue] As discussed in response to comment #10f above, the previous works had to make specific hypothesis about the "new clusters" and then find evidence to support the hypothesis. Such approach can be only done in an "explorative" mode but cannot be generalized for a benchmark study. As we discussed in #10f above, if the hypothesis is wrong, we cannot tell if the cluster is false positive or if we need a new hypothesis.

We added more clarification about co-embedding from above into the "additional discussion" section.

*10h. The authors stated: "In this approach the scHi-C and the paired data type will affect each other's embedding: the clustering of scHi-C will be certainly improved, but the clustering of paired RNA or DNA methylation modality can be either better or worse in the co-embedding."
=> This just does not make sense. The goal of cell embedding is to reveal cell states, which is the underlying truth, regardless of modality. This is a question to be explored with such co-assayed data.*

Embedding is the representation of single cell data. Single cells are placed in the embedding space.

If we do cell clustering only with the scRNA or scMethyl-C modality of the co-assay data, the cell clusters are typically very good which we used as ground truth in this study. However, if we include scHi-C data for clustering, sometimes it may reveal more cell states as the reviewer wished. However, the noises and biases inherited from the scHi-C data may negatively impact the embedding depending on how the scHi-C embedding is done.

We added more clarification about co-embedding from above into the "additional discussion" section.

*10i. The authors argued that "it will circular reasoning because the paired modality is included in the embedding."
=> Again, this is not circular if the goal is to assess cluster quality and reveal unique features by collectively considering scHi-C and other modalities.*

[scope issue] As discussed in response to comment #1,#4,#9 and #10f above, we don't have a ground truth to benchmark the cell clusters from co-embedding paired data. The approaches mentioned by the reviewer cannot be used for benchmark analysis. In co-embedding analysis, it will be circular if we use cell labeling from the paired scMethylC or scRNA-seq modality as ground truth.

We added more clarification about co-embedding from above into the "additional discussion" section.

10j. The authors did not respond to the previous comment #6: "Methods like the recent version of Fast-Higashi that allows joint embedding of multi-resolution scHi-C data would perform better."

The Fash-Higashi paper did not report the multi-resolution option and there are no documentations or examples in fast-Higashi GitHub on how multi-resolution joint embedding works or should be implemented. Furthermore, because fast-Higashi already has out-of-memory issue at 100kb resolution alone (under default options, see our responses to comment #7 & #8 above), we expect to have the same issue if 100kb resolution is included into a multi-resolution embedding.

10k. The authors stated: “We do not recommend Fast-Higashi because it is more memory demanding than Higashi and does not yield better results than Higashi or SnapATAC2.”

=> Again, this statement does not hold based on our tests. Performance differences vary depending on the dataset, which was also shown in the SnapATAC2 work.

We are not sure what issues the reviewer met with SnapATAC2. If the issues are related to the identification of rare cell types or subpopulations, that will be consistent with our findings. We showed in **Fig.6** that SnapATAC2 does not work very well distinguishing neuron subtypes even at high resolution.

10l. The authors did not carefully address previous comment #7, which remains one of the most critical concerns. The author stated: “Most importantly, we do not have a ground truth to benchmark a tool’s ability to discover new cell populations. New cell types or subpopulations from single-cell data require orthogonal experimental evidence. As a benchmark paper, we must choose the most reliable cell type annotation.”

=> As elaborated above, this is not correct. One can look into cells with distinct clusters/groups to assess whether new 3D genome features can be identified. This, indeed, is the major advantage that embeddings can offer, but unfortunately, the authors focus only on ARI scores rather than evaluating the new biological insights that can be derived from such methods.

[scope issue] As discussed in response to comment #1, #4, #9, #10f above, the reviewer’s approach had to make specific hypothesis about the “distinct clusters” and then find evidence, such as marker gene expression, to support the hypothesis. This approach can be only done in an “explorative” mode but cannot be generalized for a benchmark study. If the hypothesis is wrong, we cannot tell if the new cluster is false positive or if we just need a new hypothesis. It is therefore not possible to control the false positives, and we cannot do benchmark with this approach.

We added more clarification about co-embedding from above into the “additional discussion” section.

In conclusion, while the authors have added some evaluations and addressed a few points, the major concerns raised in the previous review remain unresolved. The limited scope, the lack of rigor and transparency of the evaluation, and the lack of biological contextualization of the evaluation in this study significantly undermines its potential utility of this benchmark study and its impact.

We appreciate the comments from the reviewer but unfortunately must reiterate that without ground truth, it is not feasible to add new benchmarks about the “downstream tasks” of scHi-C analyses, such as “new cell type discovery” or “single cell imputation”. We have detailed the reasons in our responses comment #1, #4, #9, #10f above. Notably, although the reviewer argues that it is possible to verify some of discoveries through “more careful ways”, those ways cannot support a rigorous benchmark study due to the lack of ability to determine false discoveries.

We want to emphasize that the significance of study is to benchmark embedding seeking the best representation of scHi-C data, including the resolution, loop size, and various preprocessing steps. Existing tools use very different options representing the scHi-C data, and our focus is to evaluate the impact of these options which is missing from current literature. Optimal embedding has fundamental importance and conclusions from our study applies all to all “downstream tasks”.

We have addressed the other criticisms in the point-by-point above.

Reviewer #1 (Remarks on code availability):

The code does not provide all the details.

We have made a lot of efforts during the last revision to provide as much detail as we could. Along with more detailed documentation, we added 5 full tutorials for reproducing our results which can be run for free on google colab and provided downloads for every preprocessed dataset used.

We have added a page of hyperparameter usage on the GitHub:
<https://github.com/JinLabBioinfo/SCORE/tree/main/tutorials>

Reviewer #2 (Remarks to the Author):

Overall, you have done a very good job of responding to my questions and suggestions. My only minor suggestion left is that the fact that you train different Va3DE models to handle different resolutions should be stated more clearly in the manuscript, as this point was not clear to me when I read it the first time.

We thank the reviewer for the time and support.

Yes, we need to train different Va3DE models to handle different resolutions. It is the same for all other deep learning tools, including Higashi and scVI-3D.

Reviewer #4 (Remarks to the Author):

The authors have addressed our comments. I appreciated the updated GitHub website which seems quite comprehensive.

Reviewer #4 (Remarks on code availability):

Only briefly. I have not tested the Github site, but I assume the authors checked the scripts are all documented and runnable.

We thank the reviewer for the time and support.

We very much thank Reviewer #2 for the support and the careful reviewing of our revision and our point-to-point responses to the other review. Review #2 agreed that we have appropriately responded to the previous comments and made a few minor suggestions. Please see our responses to these comments below.

Reviewer #2 (Remarks to the Author):

Reviewer #1: "For the embedding evaluation itself, as pointed out in the previous review, the authors did not look closely at the results by simply relying on cell type annotations from other modalities as "ground truth". This is hardly what actual analysis would desire, and the authors did not address this."

There appears to be some misunderstanding going on here. I honestly am not sure I understand what the reviewer is suggesting ("Embedding methods for scHi-C are useful, but they need to be placed in the context of datasets and more meaningful analysis goals."). The authors seem to interpret this as the reviewer asking them to concatenate the two modalities in the embedding space ("In the co-embedding, the scHi-C component will be appended to the co-assayed scRNAseq or scMethyl-C data for each cell."), but this makes no sense to me.

In general, I think Fig. 1a already includes all possible datasets with reliable cell type identifiers. In particular, I think the co-assay datasets are a very reliable source. If the reviewer thinks the ground truth is problematic, then I think the onus is on them to make specific suggestions and some alternative ground truth.

We appreciate the comments from reviewer #2 and agree that this is outside the context of this benchmark. Our understanding is that reviewer #1 wanted us to identify new cell populations after concatenating two modalities (scHi-C/scRNA-seq or scHi-C/scMethyl-C), which shall be cell populations that cannot be discovered using the scRNA-seq or scMethyl-C data alone. We discussed previously that we cannot benchmark such "function" due to the lack of ground truth and reviewer #2 share the same opinion.

We also thank reviewer #2 for verifying the validity of the reliable cell type annotations used in our benchmark.

Reviewer #1: "However, the manuscript tends to make broad generalizations without controlling and recognizing the limitation of the dataset being used and the confounding from the variation of scHi-C data/technology itself."

I agree this is a valid concern, but I don't see any reliable way to eliminate such technical variation. In the absence of such a method, using as many datasets as possible is a good way to eliminate the bias.

We agree with reviewer #2 that this is a valid question for future work as more scHi-C datasets are collected, new protocols are proposed, and existing protocols are improved. In the first paragraph of discussion, we have added a few sentences to acknowledge the possibility of technology biases and suggest future works:

"...Since technology biases may impact the benchmark results, it is plausible that including datasets from more sources would lead to more robust conclusions. Therefore, in this study we chose ten scHi-C datasets across different protocols with reliable cell type annotations for benchmark analysis" ...

"We also suggest that as more scHi-C datasets are collected, and new protocols are proposed, some conclusions of this benchmark should be further tested to find the optimal data representations for more scHi-C applications."

Reviewer #1: "The premise of this evaluation is questionable given its narrow scope and limited impact. ..."

This point is reasonable, though again the reviewer would be on stronger ground if they made more concrete suggestions. I would suggest that comparing pseudo-bulk vs. bulk Hi-C might be a useful additional way to evaluate clustered cell types.

We thank Reviewer #2 for recognizing the practical difficulty to properly address reviewer #1's criticism. We also appreciate the suggestion about comparing to bulk Hi-C. However, for most of the biological settings we benchmarked in this study, such as the neuron subtypes in complex brain tissues, there does not exist bulk Hi-C to compare. Resolving cell populations from complex tissues is one of the major reasons for performing single-cell Hi-C (because it is not possible to generate bulk Hi-C for every cell population in a complex tissue). Bulk Hi-C data is available for the cell lines in sciHi-C datasets which profiling various mixtures of cell lines ("synthetic mixture" datasets), but most methods cluster those cell lines perfectly and therefore pseudo-bulk data will be identical across all embedding tools.

Reviewer #1: "2. The authors used several different types of datasets, some older and some more recent...."

I agree that the conclusions are somewhat "over-claimed" here. But considering the limited availability of scHiC datasets, this may be the best the analysis one can do. It is almost impossible to remove such technical biases. In their response, the authors stood by their original claims, but I think they would do better to simply weaken their conclusions here somewhat, admitting more directly the limitations of any benchmarking study in this field.

Again, we appreciate the suggestion by reviewer #2. Following the reviewer's suggestion and as mentioned above, in the first paragraph of discussion, we now acknowledge the possibility that technology biases may impact on the benchmark results:

"...Since technology biases may impact the benchmark results, it is plausible that including datasets from more sources would lead to more robust conclusions. Therefore, in this study we chose ten scHi-C datasets across different protocols with reliable cell type annotations for benchmark analysis"...

"We also suggest that as more scHi-C datasets are collected, and new protocols are proposed, some conclusions of this benchmark should be further tested to find the optimal data representations for more scHi-C applications."

Reviewer #1: "3. The evaluation remains simplified to some clustering-based metrics, in particular ARI score. ..."

This is a valid concern about over-reliance on the ARI measure. The study would be more convincing if additional performance measures were included in Fig 2-6. Normalized mutual information, Fawkes-Mallows index, purity, homogeneity, completeness, and the V-measure are all ways to get a more nuanced view of the performance. I'm not saying you need to compute all of these, but careful thought to what measures give a complete picture of the performance of the clustering would be beneficial.

We agree with the reviewer and note that in previous rounds of reviews, other reviewers requested additional clustering metrics besides ARI. In response to those comments, we have already reported in Supp Table 2 including NMI, ASW, and AvgBIO.

Reviewer #1: "4. In the previous round of review, I raised the concern about whether the cell types defined by RNA or methylation can be treated as "ground truth". ..."

This concern is valid; however, personally I think the other modality's clustering is

typically more reliable than scHiC since their resolution is much higher than scHiC. In general, it would of course be nice if the authors could explore the scHiC embeddings more carefully, but I don't know what such an analysis would look like and Reviewer #1 didn't make any specific suggestions.

We again thank reviewer #2 for recognizing the practical difficulties in properly responding to this criticism. As we explained in our previous point-by-point document, without ground-truth cell type annotations it would not be possible to perform a reliable comprehensive benchmark analysis.

Reviewer #1: "5. There are a lot of details that are missing, ..."

This was a valid concern, which I think the authors have now addressed.

Reviewer #1: "6. The hyperparameter settings used by the authors for their own method and other baseline methods were not explained. ..."

Again, these are valid concerns which have been well addressed in the revision.

We appreciate the reviewer looking through our detailed additions to the paper. We have clarified the missing details that the initial reviewer pointed out and it has improved the clarity of this work.

Reviewer #1: "7. The claim that methods such as scHiCluster and Fast-Higashi need to construct the entire $N \times M \times M$ matrices and hence lead to out-of-memory error is incorrect. ...

8. On our end, both scHiCluster and Fast-Higashi can run on the complete human PFC dataset (4k cells) at 100Kb resolution consuming less than 40GB of memory with recommended parameters. "

The authors appear to have addressed this concern.

We thank the reviewer for looking into the details of this. Indeed, we did need to clarify the certain hyperparameters we were using such as *Fast-Higashi* with random-walk enabled, and full-matrix *scHiCluster* as originally published. With these confirmed, our initial memory analysis remains the same.

Reviewer #1: "9. The last section on integration based on scGAD is very weak. ..."

At this point, the reviewer re-raises the point about co-assays, and here they more clearly state that they want the authors "to expand the evaluation on co-embedding with such paired co-assay data with scHi-C." I think the authors have already used the co-assay data in the right way, namely, as a source of orthogonal gold standard labels of cell types. Further asking them to actually do co-embedding of the two data types doesn't really make sense to me, since as the authors correctly point out, you then don't have any gold standard. More generally, I anticipate that developing a co-embedding method for scHi-C and other, linear assay would be quite challenging and is clearly beyond the scope of this paper.

We thank the reviewer for pointing this out regarding co-embedding. Indeed, we agree that it is beyond the scope of this paper although we are also excited about future work in this direction,

Reviewer #1: "10. I read the authors' responses carefully. Here are my responses. ..."

I think the authors have responded well to all of these points, with the exception of point d. It's certainly possible to do downsampling to benchmark imputation. However, I don't think adding this experiment would add much value to the paper.

We appreciate the reviewer's comments and agree that although this is possible to do down-sampling to benchmark imputation, adding this would not add much value to the paper as it is beyond the scope of our benchmark.